# Graph Neural Networks Are More Than Filters: Revisiting and Benchmarking from A Spectral Perspective

**Yushun Dong[†], Patrick Soga[‡], Yinhan He[‡], Song Wang[‡], Jundong Li[‡]**
[†]Florida State University
[‡]University of Virginia
yushun.dong@fsu.edu, {zqe3cg,nee7ne,sw3wv,jundong}@virginia.edu

## ABSTRACT

Graph Neural Networks (GNNs) have achieved remarkable success in various graph-based learning tasks. While their performance is often attributed to the powerful neighborhood aggregation mechanism, recent studies suggest that other components such as non-linear layers may also significantly affecting how GNNs process the input graph data in the spectral domain. Such evidence challenges the prevalent opinion that neighborhood aggregation mechanisms dominate the behavioral characteristics of GNNs in the spectral domain. To demystify such a conflict, this paper introduces a comprehensive benchmark to measure and evaluate GNNs' capability in capturing and leveraging the information encoded in different frequency components of the input graph data. Specifically, we first conduct an exploratory study demonstrating that GNNs can flexibly yield outputs with diverse frequency components even when certain frequencies are absent or filtered out from the input graph data. We then formulate a novel research problem of measuring and benchmarking the performance of GNNs from a spectral perspective. To take an initial step towards a comprehensive benchmark, we design an evaluation protocol supported by comprehensive theoretical analysis. Finally, we introduce a comprehensive benchmark on real-world datasets, revealing insights that challenge prevalent opinions from a spectral perspective. We believe that our findings will open new avenues for future advancements in this area. Our implementations can be found at: `https://github.com/yushundong/Spectral-benchmark`.

## 1 INTRODUCTION

Graph Neural Networks (GNNs) have shown remarkable performances in modeling graphs in a plethora of domains, such as social media analysis (Fan et al., 2019; Ying et al., 2018), molecular biology (Wang et al., 2022; Liu et al., 2022; Gasteiger et al., 2021), and cybersecurity (Jin et al., 2020; Zhang & Zitnik, 2020; Tang et al., 2022), to name a few. The huge success of GNNs is generally attributed to its powerful neighborhood aggregation mechanism (Xu et al., 2018; Zhu et al., 2020). Specifically, such a mechanism allows each node to contribute key information to its neighbors across the graph topology (Liu & Zhou, 2022), which enables GNNs to learn informative representations and perform accurate predictions in graph-based learning tasks (Wu et al., 2022a).

To gain a deeper understanding of the reason why such a neighborhood aggregation mechanism brings revolutionary performance improvement, recent years have witnessed a huge amount of explorations (Jegelka, 2022; Xu et al., 2018). Currently, a widely acknowledged belief is that neighborhood aggregation mechanism acts as a graph signal filter (Luan et al., 2024), which serves as the dominant module in GNNs (Wang & Zhang, 2022; Bianchi et al., 2021). In most traditional GNNs, such a mechanism acts as a low-passing filter (Chang et al., 2021; Nt & Maehara, 2019) to capture the frequency components encoded with the most task-relevant information in most graph datasets. More recent studies such as (Bo et al., 2021; Bianchi et al., 2021; Luan et al., 2022), have noticed that the most task-relevant information is not necessarily encoded in the lowest frequencies, e.g., in heterophilous graphs (Zheng et al., 2022; Zhu et al., 2021). To facilitate more capable GNNs to han-

dle different types of graphs, a variety of complicated neighborhood aggregation mechanisms have been designed to equip GNNs with more flexible filters (Li et al., 2018; Guo et al., 2023), aiming at capturing more task-relevant information from different frequency components across the spectral domain. Nevertheless, a significant flaw arises when we zoom in on these works. Specifically, the motivation of designing more flexible filters is implicitly based on the assumption that *it is difficult for GNNs to yield outputs with abundant frequency components if these components are significantly weakened or filtered out by the neighborhood aggregation mechanism*. However, GNNs are more than filters associated with neighborhood aggregation mechanisms. Other modules, e.g., non-linear layers, are often included as well. This fact naturally undermines the validity of this assumption. Considering such a gap, in this work, we ask:

> *When such filters are fixed, can GNNs still flexibly output different frequency components?*

The above question is critical since its answer determines whether we should attribute most strengths and weaknesses of GNNs from a spectral perspective to such filters or not. Despite the scarcity of existing explorations, several studies have observed that non-linear layers can affect the frequency components of the GNNs' output (Balcilar et al., 2021b; Yang et al., 2024). These results imply that GNNs as a whole may exhibit different behavioral characteristics, e.g., the GNNs' output frequency components given a certain input frequency component, compared with the above-mentioned neighborhood aggregation mechanisms in the spectral domain. However, most existing works fail to gain a comprehensive understanding of how the components other than those filters influence the behavior of GNNs in the spectral domain and finally affect the performance in graph learning tasks.

We note that it is non-trivial to properly answer the aforementioned question. In particular, we mainly face three fundamental challenges. (1) **Complex Analytical Expressions.** GNNs are usually highly complex when the components other than neighborhood aggregation mechanisms are considered all together. Therefore, its analytical expression can hardly be exploited to perform analysis in the spectral domain. (2) **Lack of Frequency-Specific Incentives.** GNNs are typically supervised with a set of fixed ground-truth labels during training. However, these ground-truth labels are usually a composition of different frequency components and do not show clear incentive patterns preferring certain frequency components. Therefore, it becomes difficult to analyze whether GNNs are able to yield outputs with components that they have not previously observed. (3) **Lack of Metric and Benchmark.** To the best of our knowledge, there is currently no metric that measures the flexibility of a GNN's output frequency components in the spectral domain. On the other hand, it is also necessary for us to understand the differences between different popular GNN models on the question above. However, no existing benchmark can comprehensively reveal such insights.

In this paper, we take an initial step to investigate the problems above. Specifically, we first perform an exploratory study, where we avoid characterizing any analytical expressions to tackle our first challenge. Instead, we propose to design appropriate supervision information as the frequency-specific incentives for GNN training, such that the capacity of GNNs in yielding different frequency components can be exposed and observed across the frequency axis. This helps us to properly tackle the second challenge. The observations of the exploratory study verify that the modules other than neighborhood aggregation can already enable GNNs to flexibly output different frequency components at different energy levels, which challenges the prevalent opinion that neighborhood aggregation typically dominates GNNs' behavioral characteristics in the spectral domain. Furthermore, we design evaluation protocols to evaluate the performance of different GNNs in yielding different frequency components across the spectral domain. We finally present a comprehensive benchmark under such protocols to introduce a comprehensive understanding on such a problem across different popular GNNs, which properly handles the third challenge. We summarize our contributions below:

- **An Exploratory Study Challenging the Common Belief.** We propose a principled strategy to characterize the flexibility of GNNs to yield different frequency components. Surprisingly, we found that the filter resulted from the neighborhood aggregation does not dominate the behavioral characteristics of GNNs in the spectral domain, which disagrees with the prevalent opinion.

- **A Novel Research Problem.** We formulate a novel research problem of *Measuring and Benchmarking the Performance of GNNs From A Spectral Perspective* and take an initial step towards properly handling it. We provide valuable insights through explorations towards understanding the behavioral characteristics of GNNs in the spectral domain.

- **A Comprehensive Benchmark with Novel Evaluation Protocol.** We design a novel evaluation protocol with solid theoretical basis and practical significance. We also conduct extensive experi-

ments to enhance understanding of popular GNNs on real-world datasets. Notably, our benchmark provides a consistent view directly applicable to both spatial- and spectral-based GNNs.

## 2  PRELIMINARIES

**Notations.** Throughout our work, without further specification, italic letters (e.g., $\mathcal{X}$), bold uppercase letters (e.g., $\mathbf{X}$), bold lowercase letters (e.g., $\mathbf{x}$), and ordinary lowercase letters (e.g., $x$) represent matrices, vectors, and scalars, respectively. For any matrix, e.g., $\mathbf{X}$, we employ $\mathbf{X}_i$ and $\mathbf{X}^i$ to indicate its $i$-th row and column, respectively. We denote an undirected graph as $G = (\mathcal{V}, \mathcal{E})$, where $\mathcal{V} = \{v_1, ..., v_n\}$ and $\mathcal{E}$ are the set of nodes and edges. We denote $\mathbf{A} \in \mathbb{R}^{N \times N}$ as the graph adjacency matrix in which $\mathbf{A}_{i,j} = 1$ if there exists an edge between node $i$ and node $j$, otherwise $\mathbf{A}_{i,j} = 0$. With graph adjacency matrix $\mathbf{A}$, the graph node degree matrix can be defined as $\mathbf{D} = diag(d_1, ..., d_N)$, where $d_i = \Sigma_j \mathbf{A}_{i,j}$. The normalized graph Laplacian matrix is defined as $\mathbf{L} = \mathbf{I} - \mathbf{D}^{-\frac{1}{2}} \mathbf{A} \mathbf{D}^{-\frac{1}{2}}$. Additionally, we define the graph node feature matrix $\mathbf{X} \in \mathbb{R}^{N \times F}$, where $\mathbf{X}^j$ represents the $j$-th feature channel and $F$ denotes the number of feature channels. We employ the sign $\odot$ as the Hadamard multiplication.

**Current Progress of Gnns & Concerns From a Spectral Perspective.** There are two mainstream lines of research on GNNs, i.e., spatial- and spectral-based ones. Researchers examining the spatial perspective consider the aggregation process of GNNs as a node attribute aggregator across the graph topology (Kipf & Welling, 2017; Wu et al., 2020; Xu et al., 2018). In contrast, those exploring GNNs from a spectral perspective consider the aggregation process as a filter in the spectral domain, i.e., the eigenspace of the graph Laplacian matrix $\mathbf{L}$, largely considering GNNs as low-passing filters (Nt & Maehara, 2019; Chang et al., 2021; Yu & Qin, 2020). Recently, diverse designs of filters associated with different neighborhood aggregation mechanisms have been proposed to help GNNs capture the key information encoded in different frequency components (Bo et al., 2021; Guo et al., 2023; Dong et al., 2021). In general, these explorations lay a solid mathematical foundation for GNN with spectral-based methods. However, most current methods focus on the filters associated with the neighborhood aggregation mechanisms rather than considering a GNN as a whole. As such, the role of other modules such as non-linear layers in GNNs has been long neglected. In recent years, several works (Balcilar et al., 2021b; Yang et al., 2024) have provided primary evidence that the non-linear layers can effectively shift the behavioral characteristics of GNNs. To further bridge the aforementioned gap, we now formally introduce the problem of *Measuring and Benchmarking the Performance of GNNs From A Spectral Perspective* below.

**Problem 2.1. Measuring and Benchmarking the Performance of GNNs From A Spectral Perspective.** *Our goal is to qualitatively understand and quantitatively compare the capabilities of GNNs in capturing the key information encoded in different frequency components of input graphs to perform graph learning tasks.*

## 3  AN EXPLORATORY STUDY

To properly handle Problem 2.1, the prevalent strategy is to simply analyze the frequency response function associated with the filter resulted from the neighborhood aggregation mechanism (Nt & Maehara, 2019; Wu et al., 2019). However, such a straightforward approach may not be able to handle Problem 2.1, since the behavioral characteristics of GNNs in the spectral domain may not be fully dominated by such filters. Below, we show our preliminary explorations to further clarify such a common misunderstanding.

**Research Question.** Here, we perform preliminary studies to explore whether the behavioral characteristics of GNNs in the spectral domain are dominated by the filters associated with the neighborhood aggregation mechanism (as discussed in Section 1). Specifically, we are particularly interested in revealing whether GNNs can flexibly yield outputs with abundant frequency components that have been significantly weakened or filtered out by the non-learnable filter.

**Evaluation Protocol.** In this study, we measure the influence of different frequency components using the commonly adopted notion of *Energy* (Yang et al., 2022; Tang et al., 2022). We construct our experimental datasets based on real-world graph datasets. First, we compute the normalized Laplacian eigenvectors of the graph and sort them by eigenvalue (i.e., frequency). We then bin these eigenvectors into even-width bins, where each bin is associated with the mean of the eigenvectors

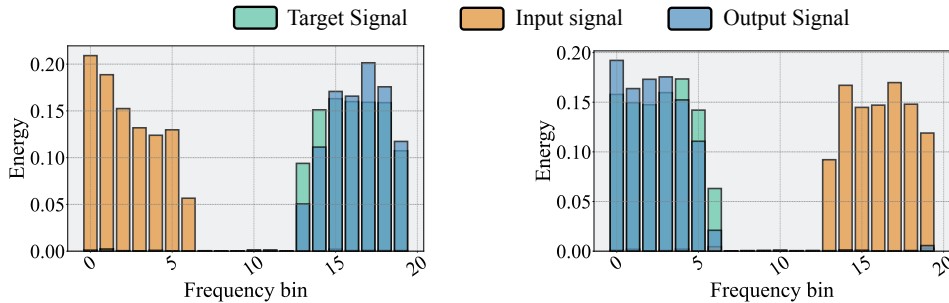

(a) Low-frequency input; high-frequency targets.  (b) High-frequency input; low-frequency targets.

Figure 1: A comparison between the energy of input and output frequency components of GCN on *Co-author CS* dataset. The results show that the output frequency components can always flexibly align with the target distribution in both cases of (a) inputting low-frequency components only but aiming to output high-frequency components; and (b) inputting high-frequency components only but aiming to output low-frequency components.

falling into that bin. The bottom, middle, and upper one third of bins are designated as low-, middle-, and high-frequency components. We propose to conduct node-level regression tasks to answer our research question above. Specifically, we set the target as the mean of the eigenvectors coming from one frequency component, while the input features are the mean of eigenvectors in a different frequency range (e.g., low-frequency eigenvectors when the target is high-frequency). In this way, the input graph signal will have zero energy levels for the frequency components of the prediction target, while the ground truth can serve as frequency-specific incentives. This simulates the scenarios where *certain frequency components are absent in the input or filtered out by the neighborhood aggregation mechanism*. We note that graph filters cannot generate frequency components that are not originally contained in the input graph data. Therefore, if a GNN model can still produce outputs with abundant target frequency components (in terms of energy levels) under various conditions, it demonstrates that the neighborhood aggregation mechanism does not necessarily dominate the output frequency components. In other words, GNNs can recover missing frequency components if it benefits the optimization goal during training, even if these components are/become absent in the forwarding process. We adopt GCN as our GNN model and conduct experiments on five real-world datasets. We train our models using MSE loss after standardizing the input features. We show two exemplary cases based on *Co-author CS* dataset in Figure 1, where the energy distribution of the final GNN's output signal alongside those of the input and target signals are visualized. We present complete results in the Appendix.

**Observations & Analysis.** Surprisingly, our experiments reveal that GNNs demonstrate remarkable capability to recover frequency components across diverse scenarios. As shown in Figure 1, even when certain frequency components are filtered out or significantly weakened in the input, GNNs can still flexibly align their output with the target distribution. This is evident in both cases: (a) where low-frequency inputs generate high-frequency outputs, and (b) where high-frequency inputs produce low-frequency outputs. Notably, this indicates that other modules, such as the non-linear layers, in GNNs can also play a crucial role in altering the energy distributions across different frequency components. Our preliminary conclusion is that the behavior of GCN in the spectral domain is not necessarily dominated by the spectral characteristics of its neighborhood aggregator. Instead, GCN can easily output different frequency components when appropriate frequency-specific incentives exist in the supervision signal. Such flexibility in yielding various frequency components is even comparable to those state-of-the-art GNNs with carefully designed learnable filters (Bo et al., 2021), and we show a more comprehensive comparison in Appendix C.

To summarize, we conclude that the frequency response of the filter of a neighborhood aggregator does not necessarily determine the behavioral characteristics of its host GNN in the frequency domain. In fact, GNNs as a whole have the capability to significantly modify their output energy distribution compared with the input energy distribution in the spectral domain. Such an insight suggests that only analyzing the flexibility of the filters associated with the neighborhood aggregation mechanism is insufficient to fully explain the strengths and weaknesses of GNNs, which is in conflict with the traditional belief. Taking a step further, a critical question emerges: How can we effectively measure the general "capturing and altering" capability of GNNs in the frequency domain? This question becomes paramount for understanding the true potential and limitations of current

GNNs. We address this question in the following section where we aim to provide a comprehensive evaluation protocol for benchmarking GNN performance from a spectral perspective.

## 4 BENCHMARK DESIGN

In this section, we introduce the benchmark design. Specifically, we first introduce the evaluation protocol, followed by comprehensive analysis to lay a solid theoretical foundation which will directly support the practical significance of our proposed benchmark from a spectral perspective.

### 4.1 EXPERIMENTAL PROTOCOL

**Downstream Task & Dataset Preparations.** While we used node regression for our preliminary study in Section 3, we adopt node classification for our extensive empirical benchmark considering its superior practical significance in graph learning tasks. Specifically, we propose to adopt the same approach to generate continuous values corresponding to each node in the input graph (as in Section 3) followed by an additional discretization process by giving thresholds to determine the ground truth labels for each node. We show in Section 4.2 that the additional discretization process only brings an upper-bounded energy distribution deviation compared with the continuous ground truth values in the node-level regression task, which ensures satisfying consistency. Meanwhile, such an approach ensures that the targets to predict possess the frequency-specific incentives needed to evaluate performance on each bin in the frequency domain. We propose to adopt real-world datasets such that we will perform evaluations on the node attributes and graph topology that bear practical significance across different domains. We further provide more details in Appendix A.

**GNNs for Benchmarking.** We conduct evaluation on a total number of 14 GNNs, namely SAGE (Hamilton et al., 2017), GCN (Kipf & Welling, 2017), GCNII (Ming Chen et al., 2020), GAT (Veličković et al., 2018), GATv2 (Brody et al., 2022), SGC (Wu et al., 2019), FA (Bo et al., 2021), GIN (Xu et al., 2018), ChebNet (Defferrard et al., 2016), GatedGraph (Li et al., 2016), the Transformer (Vaswani et al., 2017), GPS (Rampášek et al., 2022), APPNP (Gasteiger et al., 2019), and the 1-WL operator from Morris et al. (2019) (denoted as 1-GNN). These models cover a wide range of popular and state-of-the-art GNNs designed in either spatial or spectral domain.

**Real-World Datasets.** We benchmark GNNs on the full versions of the Cora and DBLP citation graphs (Bojchevski & Günnemann, 2018), the CS and Physics coauthor datasets by Shchur et al. (2018), and the Amazon-Computers and Amazon-Photo product graphs by (Shchur et al., 2018). These datasets vary in size, structure, spectral energy distribution, and semantic domains, allowing us to evaluate the GNNs' performance across diverse graph types and application areas.

**Benchmark Evaluation Metrics.** Following most other works, we adopt node classification accuracy as the primary metric to measure the performance of GNNs. On the basis of this, we now introduce the qualitative and quantitative performance evaluation methods. From a qualitative perspective, we propose the notion of *Accuracy Curve in the Spectral Domain*. Specifically, as introduced in Section 3, each round of experiments is associated with a bin on the spectral axis, based on which the node-level prediction target (a discrete label for node classification) is generated. When the input graph signal contains all frequency components at the same energy level, the performance under this bin generally reflects the capability of such a GNN model in capturing and leveraging the information encoded in the associated frequency component to perform prediction. Accordingly, the performance across all available bins on the spectral axis form a curve, which generally reflects the tendency of how such capability changes w.r.t. the frequency value. From a quantitative perspective, we propose to utilize the *Normalized Area Under the Accuracy Curve*, Normalized AUAC, to measure the general capability of each GNN model in capturing and leveraging the information encoded in different frequency components. Specifically, it is calculated as the division between the AUAC under the full frequency range and the largest possible AUAC under the full frequency range. Additionally, we are also interested in the capability of GNNs in capturing the information encoded in a certain range of frequency components. In this case, Normalized AUAC can also be adopted by specifying a particular range of frequencies.

**Implementation Details.** All experiments were conducted using PyTorch (Paszke et al., 2017) and PyTorch Geometric (Fey & Lenssen, 2019) libraries. We used 2-layer GNNs with a hidden dimension of 64 for all runs. GNNs were trained for 500 epochs using the Adam optimizer with

a learning rate of 0.001. We bin each frequency component with a width of 0.1. No learning rate scheduler or early stopping was used. For each dataset and spectral bin, we report the mean and standard deviation of the results across 3 runs. More implementation details are in Appendix A.

## 4.2 THEORETICAL ANALYSIS

We adopt node classification task for our main benchmark considering that it typically fosters a stronger practical significance. To ensure the consistency between node-level regression task and node classification tasks, in this section, we aim to reveal that the additional discretization process in node classification does not bring significant deviation from the ground truth's energy distribution in the spectral domain. Below we present the theoretical analysis revealing such insights.

We refer to the matrix of one-hot vectors representing the ground truth node class labels after discritization as the *Node Class Label* (NCL) matrix for convenience. Below we first define the *Energy Distribution Field* (EDF) of a node class label distribution.

**Definition 4.1.** (Energy Distribution Field) The energy distribution field, denoted as $\mathcal{F}_{\mathbf{M}}$, of an NCL matrix $\mathbf{M} \in \mathbb{R}^{n \times k}$ is the set of energy distributions of the unit vectors whose corresponding NCL matrix is $\mathbf{M}$. In other words, $\mathcal{F}_{\mathbf{M}} := \{e(v) \in \mathbb{R}^{n \times 1} | ||v||_2 = 1, \tau(v) = \mathbf{M}\}$, where $e(\cdot)$ is the energy distribution function mapping a vector to its energy distribution in the graph spectrum field, $\tau(\cdot)$ is the function mapping unit vectors to their NCL matrices.

With the concept of EDF, we then formulate the "closeness" of the energy distribution between the eigenvector $v$ of the graph Laplacian and its corresponding NCL matrix $\tau(v)$ as $\max_{e_u \in \mathcal{F}_{\tau(v)}} ||e_u - e(v)||_2$. We take two steps to verify that the optimal value of the maximization problem is small enough: *(i)* We prove that the energy distribution function is Lipschitz, which indicates that the energy distribution function is a "smooth" function where quantifying its function value variations is equivalent to quantifying its variable variations; *(ii)* The inverse image of the energy distribution function $e(\cdot)$ on any energy distribution field $\mathcal{F}_{\mathbf{M}}$ is small enough.

For the first step, since the energy distribution function is essentially a linear function followed by a vector normalization procedure, we can write the energy distribution function as $e(v) = \frac{(\mathbf{A}v) \odot (\mathbf{A}v)}{||\mathbf{A}v||^2}$ with $\mathbf{A}$ being an orthonormal matrix. Actually, the energy distribution function is Lipschitz:

**Theorem 4.2.** *The energy distribution function $e(v) = \frac{(\mathbf{A}v) \odot (\mathbf{A}v)}{||\mathbf{A}v||^2}$, with $\mathbf{A}$ being orthonormal, is Lipschitz on the unit sphere.*

For the second step, we prove the inverse image of the energy distribution function $e(\cdot)$ on any EDF $\mathcal{F}_{\mathbf{M}}$ is "small enough". Intuitively, we can calculate the area of $e^{-1}(\mathcal{F}_{\mathbf{M}})$ on the unit sphere and prove that the area is "small enough". However, since the calculation of the area is rather complex for a high-dimensional sphere, we instead prove that the variation of center angle in $e^{-1}(\mathcal{F}_{\mathbf{M}})$ is small enough. Specifically, we have the following theorem.

**Theorem 4.3.** *Assume that we segment $[-1, 1]$ (i.e., the value field of any entry of an eigenvector of a graph Laplacian matrix) into $k$ intervals with identical lengths, as illustrated in Figure 6(a). Consider the Euclidean space of $[-1, 1]^n$ separated into $k^n$ identical hypercubes. We claim that each section of the unit sphere and a hypercube corresponds to a $e^{-1}(\mathcal{F}_{\mathbf{M}})$ for some NCL matrix $\mathbf{M}$. We denote the maximum center angular variation of $e^{-1}(\mathcal{F}_{\mathbf{M}})$ among all NCL matrices $\mathbf{M}$s as $\theta_{\max}$. For $\forall \epsilon \geq 0$, there exists a $k_0 = 2(\frac{2}{\pi e^{\frac{1}{e}}})^{\frac{n}{2}}$, when $k \geq k_0$, then $\theta_{\max} \leq \epsilon$.*

Now, we are able to provide an upper bound for the energy distribution variance for EDF $\mathcal{F}_{\mathbf{M}}$ of any NCL matrix $\mathbf{M}$.

**Theorem 4.4.** *Given the conditions of Theorem B.3, for any eigenvector $v$ of Laplacian amtrix of graph $\mathcal{G}$, we have $\max_{v \in \mathcal{S}^{n-1}, e_u \in \mathcal{F}_{\tau(v)}} ||e_u - e(v)||_2 \leq 2(\frac{4n}{k^2})^{\frac{1}{n}}$, where $\mathcal{S}^{n-1}$ is $n-1$ dimensional unit sphere embedded in $n$ dimensional Euclidean space.*

By Theorem B.4, we are able to reveal that the additional discretization process in node classification task does not bring significant deviation in the ground truth's energy distribution in the spectral domain. Meanwhile, given a tolerance of the change of energy distribution from an eigenvector and its corresponding NCL matrix, we can also estimate the minimum number of identical interval segments $k$ for the specification of the discretization thresholds to satisfy precision requirements.

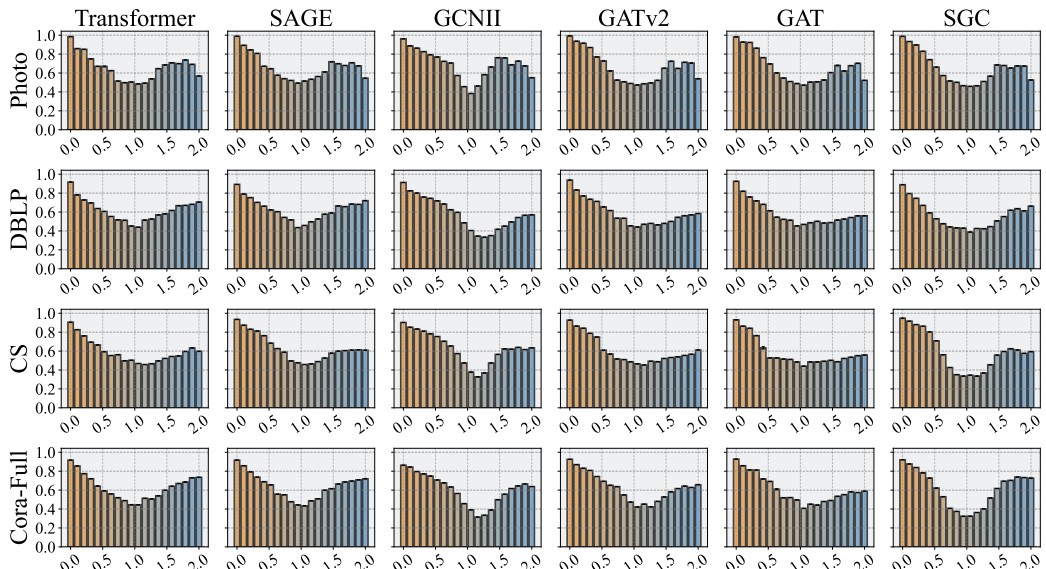

Figure 2: The accuracy curves of different GNNs in the whole spectral domain. In each subplot, the $x$-axis represents the frequency and the $y$-axis represents the accuracy of GNNs in the node classification task with the ground truth labels derived from the associated frequency bin.

## 5 EMPIRICAL INVESTIGATION

### 5.1 RESEARCH QUESTIONS

We are interested in answering four research questions (RQs) below. **RQ1**: What are the trends of GNNs' performance in capturing the information encoded in different frequency components across the frequency domain? **RQ2**: How do various GNNs compare in their ability to capture information across different frequency components, both holistically and in specific ranges? **RQ3**: How will the benchmark show its practical implications to guide a practitioner's choice of GNN? **RQ4**: How does the depth of GNNs affect the proposed accuracy curves in the spectral domain?

### 5.2 QUALITATIVE PERFORMANCE COMPARISON IN THE SPECTRAL DOMAIN

We first answer RQ1 by analyzing the general trends that GNNs show in our benchmark, and we present the general tendency of the accuracy curves in the spectral domain for different GNNs in Figure 2. We note that only partial results are presented here due to space limit, and see Appendix D for complete results. We have the following observations.

First, from the perspective of a general tendency, we observe that GNNs show V-shaped curves in all cases. Such a phenomenon indicates that GNNs typically have stronger capability in capturing the information encoded in the lowest and highest frequency components (e.g., those associate with the smallest and largest frequency values) compared with the middle frequency components. Specifically, when the task-relevant information is encoded in the low frequency components, the neighborhood aggregation can directly benefit the prediction of each node. This is because nodes with similar labels tend to connect with each other, which can significantly facilitate predictive performance. On the other hand, when the task-relevant information is encoded in the high frequency components, the neighbors of each node can also contribute to its own prediction if the GNNs are able to learn not to predict the same label for connected nodes. However, in contrast to the prevalent opinion, the capability of capturing the task-relevant information encoded in the middle frequency components is the most difficult for all adopted GNNs. A key reason for this phenomenon is that in this case, the label distribution of each node's neighborhood is generally uniform. Therefore, it becomes very difficult for GNNs to learn a general criterion to predict the label of a node based on its neighbors. Based on the discussion above, we argue that the GNNs' weakness in capturing the task-relevant information encoded in the middle frequency components reveals an inherent limitation of neighborhood aggregation mechanism.

Table 1: Quantitative results of Normalized AUAC score in percentage for 14 GNN models across six real-world datasets. The best ones are in **Bold** and the second best ones are underlined. We also mark out the average ranking for each model across all datasets.

| Model | Computers | Cora | CS | DBLP | Photo | Physics | Avg. Ranking |
|---|---|---|---|---|---|---|---|
| APPNP | $56.94 \pm 1.08$ | $54.26 \pm 1.80$ | $52.90 \pm 2.16$ | $51.69 \pm 0.40$ | $56.77 \pm 1.51$ | $53.60 \pm 1.38$ | $13.67 \pm 0.75$ |
| FA | $58.76 \pm 1.42$ | $54.88 \pm 2.82$ | $52.95 \pm 3.79$ | $52.76 \pm 1.10$ | $61.79 \pm 1.78$ | $55.81 \pm 2.33$ | $12.00 \pm 1.41$ |
| GIN | $55.39 \pm 0.86$ | $60.98 \pm 2.94$ | $57.90 \pm 3.35$ | $58.99 \pm 2.95$ | $60.57 \pm 1.48$ | $59.65 \pm 3.27$ | $10.33 \pm 2.36$ |
| SGC | $62.07 \pm 2.38$ | $61.09 \pm 3.71$ | $59.25 \pm 4.19$ | $56.36 \pm 1.95$ | $64.98 \pm 2.59$ | $58.87 \pm 3.38$ | $9.50 \pm 1.98$ |
| GAT | $63.47 \pm 2.50$ | $60.33 \pm 2.26$ | $58.28 \pm 2.09$ | $58.41 \pm 1.67$ | $65.56 \pm 2.59$ | $60.38 \pm 2.10$ | $8.67 \pm 1.89$ |
| GPS | $60.94 \pm 1.59$ | $58.23 \pm 0.99$ | $59.89 \pm 1.73$ | $58.94 \pm 0.92$ | $63.53 \pm 1.77$ | $62.90 \pm 1.52$ | $8.33 \pm 2.13$ |
| GCN | $63.42 \pm 2.39$ | $62.17 \pm 4.18$ | $59.82 \pm 4.04$ | $57.21 \pm 2.09$ | $66.20 \pm 2.78$ | $59.49 \pm 3.61$ | $7.67 \pm 2.81$ |
| GatedGraph | $58.67 \pm 1.22$ | $\mathbf{69.85 \pm 2.74}$ | $66.14 \pm 2.93$ | $63.84 \pm 2.56$ | $60.41 \pm 1.13$ | $60.20 \pm 2.86$ | $6.17 \pm 4.60$ |
| Transformer | $64.83 \pm 1.75$ | $62.89 \pm 1.75$ | $59.43 \pm 1.50$ | $61.86 \pm 1.36$ | $65.86 \pm 1.87$ | $60.93 \pm 1.49$ | $5.50 \pm 2.22$ |
| Cheb | $63.63 \pm 1.65$ | $60.54 \pm 1.40$ | $60.88 \pm 1.93$ | $61.14 \pm 1.28$ | $65.88 \pm 1.68$ | $62.95 \pm 2.06$ | $5.50 \pm 1.61$ |
| 1-GNN | $56.04 \pm 1.12$ | $\underline{68.00 \pm 2.45}$ | $\mathbf{66.27 \pm 2.77}$ | $\mathbf{64.16 \pm 1.96}$ | $58.17 \pm 1.47$ | $65.16 \pm 2.77$ | $5.33 \pm 5.44$ |
| GATv2 | $64.22 \pm 2.46$ | $63.05 \pm 2.21$ | $60.41 \pm 2.12$ | $59.38 \pm 1.92$ | $\underline{66.54 \pm 2.73}$ | $62.58 \pm 2.12$ | $4.67 \pm 1.49$ |
| GCNII | $\mathbf{66.57 \pm 2.10}$ | $60.54 \pm 2.73$ | $62.91 \pm 2.75$ | $58.13 \pm 3.00$ | $\mathbf{69.03 \pm 2.30}$ | $\mathbf{66.55 \pm 2.50}$ | $4.50 \pm 4.03$ |
| SAGE | $64.60 \pm 1.69$ | $63.97 \pm 1.80$ | $63.17 \pm 2.01$ | $62.88 \pm 1.33$ | $66.13 \pm 1.88$ | $64.29 \pm 2.10$ | $3.17 \pm 0.37$ |

Second, we are also interested in comparing tendencies across different GNNs and datasets. Specifically, we observe consistent tendencies of accuracy curves in the spectral domain for the same GNN model across different datasets, which validates the stability of our proposed evaluation strategy. Meanwhile, on the same dataset, different GNNs can show different tendencies across the frequency axis, which demonstrates the difference in the capability of GNNs to capture task-relevant information. This further reveals the significance of the proposed evaluation method in deriving a stable and reliable evaluation for the performance of different GNNs in the spectral domain.

## 5.3 QUANTITATIVE PERFORMANCE COMPARISON IN THE SPECTRAL DOMAIN

We now answer RQ2 by analyzing the quantitative results of Normalized AUAC score in percentage for 14 GNN models across six real-world datasets, and we show the results in Table 1. According to Section 4, the quantitative value of the Normalized AUAC score reflects the general capability of GNNs to capture the task-relevant information encoded in different frequency ranges. In addition, we are also interested in analyzing this capability on specific ranges of frequencies. Therefore, we split the range of frequencies evenly into three sections, namely the low frequency frange (the components with a frequency value ranks at the bottom one third), the middle frequency range (the components with a frequency value ranks at the middle one third), and the high frequency range (the components with a frequency value ranks at the top one third). We show the average rankings of GNNs by measuring their Normalized AUAC score in these three different frequency ranges above in Figure 3. Notably, the proposed benchmark is able to analyze the performance of both spatial- and spectral-based GNNs from a consistent spectral view. Therefore, we are able to involve both types of GNNs into the comparison at the same time. We have the observations below.

Comparing models focusing on the general spectral domain, we observe that the performance measured by Normalized AUAC score gives a ranking that challenges the traditional assumptions on the relative superiority of these GNNs. Specifically, SAGE, GCNII, and GATv2 are the GNNs showing the best performance in terms of the average ranking across all datasets. In fact, these GNNs are rarely discussed and evaluated from a spectral perspective. The primary reason is that they are mostly designed in the spatial domain and most do not have a well-defined frequency response function. We argue that the advantages of these GNNs in the spectral domain are largely neglected by the current spectral-based studies (Nt & Maehara, 2019; Wu et al., 2019) due to their reliance on an explicit analytical form of frequency response function.

Comparing models focusing on the three specific sections of frequency components, we observe that different GNNs specialize in extracting task-relevant information from different ranges of frequencies. Specifically, GNNs specialize in learning from low (e.g., SAGE and Cheb) and high (e.g., GPS and Transformer) frequency components typically do not show strong superiority over other GNNs in learning from middle frequency components. Instead, GNNs such as GATv2 and GCN achieve clear superiority over other GNNs in learning from middle frequency components, which is often ignored by other studies relying on frequency response analysis.

## 5.4 CASE STUDY

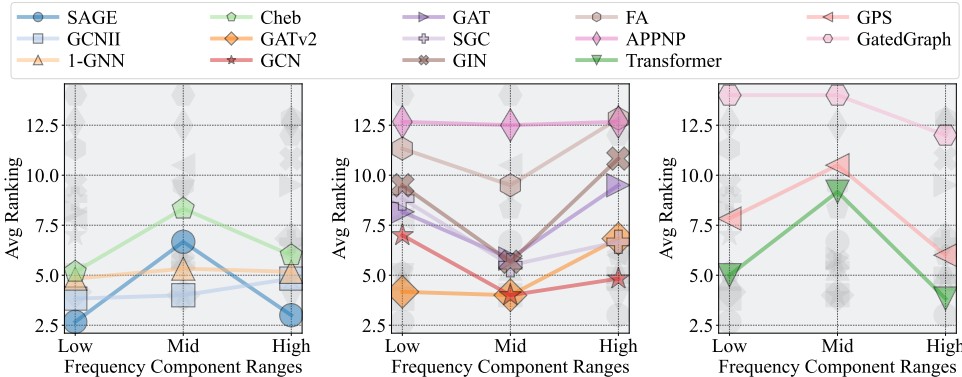

Figure 3: Performance comparison in the average ranking of 14 GNNs on six real-world datasets. The GNNs are shown by obtaining the best rankings on low frequency components (left), on middle frequency components (middle), and on high frequency components (right).

In this subsection, we conduct a case study to explore RQ3. Specifically, we propose to simulate the real-world problem of needing to choose from a variety of different GNNs, and we then analyze how the findings from our benchmark can help practitioners. Specifically, we adopt six new real-world datasets, namely Airport-Brazil (AB), Wisconsin (Wisc.), Cornell (Corn.), Squirrel (Squi.), and Chameleon (Cham.). Most of these dataests are reported to be difficult for node classification task with GNNs. We first perform node classification with all 14 GNNs based on these datasets and then derive the ranking of all GNNs as $r_1$. Here

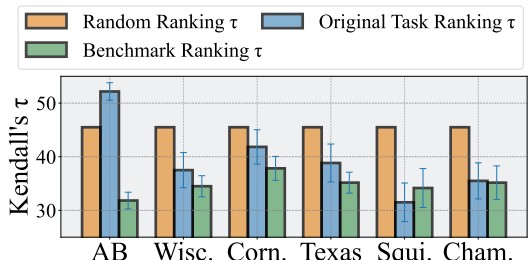

Figure 4: Kendall's $\tau$ comparison on new datasets between a random ranking (orange), the rankings from the original graph learning task (blue), and the average rankings from our benchmark (green).

$r_1$ serves as the actual ranking we seek to know to pick the best GNNs to use. However, in practice, we may not always be able to perform experiments prior to making a choice. Therefore, we propose to analyze how well other rankings can approximate such an actual ranking. We first analyze the node labels in the training node set to identify the frequency component range (low, mid, or high) where most energy falls in. According to this range, we collect the associated performance ranking $r_2$ from Section 5.3 as the ranking derived from our benchmark. As a comparison, we also collect random rankings $r_3$ and the node classification rankings $r_4$ directly derived from the node classification task on the chosen six datasets.

We show the average Kendall-Tau (KT) distance $\tau$ (total number of inversions for any two positions $i$ and $j$ where $i > j$) between the actual ranking $r_1$ and the benchmark ranking $r_2$ in Figure 4. Here, a smaller KT distance indicates larger similarity between the two rankings. We found that the KT distance between $r_1$ and $r_2$ (Benchmark Ranking $\tau$) is significantly smaller than that between $r_1$ and $r_3$ (Random Ranking $\tau$) or $r_4$ (Original Task Ranking $\tau$) in most cases, which indicates a satisfying approximation of $r_1$ with $r_2$. This reveals the practical significance of the benchmark in understanding the superiority across different GNNs prior to any experiments.

## 5.5 PARAMETER STUDY

We finally answer RQ4 by changing the depth of each GNN to explore how the results of the proposed evaluation protocol will change. We note that stacking multiple filters (by adding more iterations of neighborhood aggregation) significantly affects the frequency response. Therefore, existing studies that analyze the performance of GNNs by relying on their aggregators' frequency response functions typically have significantly different conclusions when the layer number of GNNs changes. Specifically, we range the number of GNN layers from two to four, and we present an example of the accuracy curves in the spectral domain across two-, three-, and four-layer cases in Figure 5 (see Appendix F for complete results). We observe that changing the layer number does not affect the shape of the accuracy curves in the spectral domain. Such an observation can also be found across different GNNs and datasets, which further validates the stability of the proposed evaluation protocol.

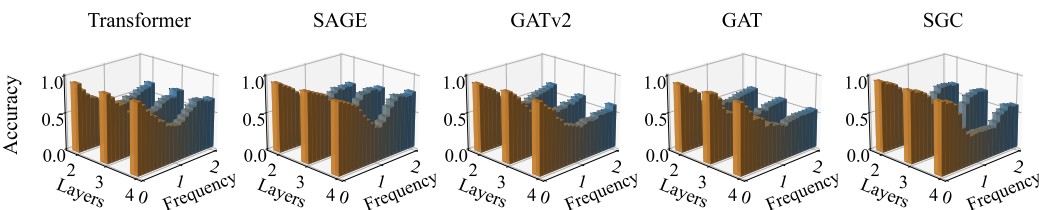

Figure 5: Accuracy curves in the spectral domain across different GNNs on the Coauthor-CS dataset. The shape of these curves does not significantly change across different layer numbers, validating the stability of the proposed evaluation protocol.

## 6 RELATED WORKS

**Spectral Graph Neural Network Analysis.** Compared to spatial GNNs (Hamilton et al., 2017), spectral GNNs typically offer a sound theoretical basis (Balcilar et al., 2021a; Bo et al., 2023). Recent studies have shown that the neighborhood aggregation mechanism in GNNs typically acts as a low-pass graph signal filter (Chang et al., 2021; Nt & Maehara, 2019), capturing the lowest frequency components that are usually relevant for various graph learning tasks. However, while more flexible filters have been proposed to better capture task-relevant information encoded in different frequency components (Li et al., 2018; Guo et al., 2023; Bianchi et al., 2021), existing research typically overlook the behavioral characteristics of GNNs in the spectral domain as a whole. As such, the roles of other modules, such as the non-linear layers, are often ignored. Recent works (Balcilar et al., 2021b; Yang et al., 2024) have show evidence that these modules other than neighborhood aggregation can significantly alter the frequency components of the GNN's output, This underscores the need for a comprehensive analysis that considers GNNs in their entirety instead of only focusing on their neighborhood aggregation mechanisms. Different from existing works, we propose to consider GNNs as a whole and conduct benchmarking form a spectral perspective.

**Graph Neural Network Benchmarking.** Research GNN benchmarks focus on evaluating the performance of GNNs in terms of accuracy (Wu et al., 2022b; Zheng et al., 2021), scalability (Duan et al., 2022), and efficiency (Baruah et al., 2021). For example, various benchmarks have been proposed to evaluate the performance of GNNs in various tasks (Dwivedi et al., 2023), such as recommendations (Wu et al., 2022a) and molecular property predictions (Fung et al., 2021). In terms of the scalability of GNNs, different benchmarks are also proposed, such as LRGB (Dwivedi et al., 2022) for long-range dependency and LS-Bench for training on large-scale graphs. The efficiency of GNNs is also studied in various benchmarking works (Gong & Kumar, 2024). Despite existing efforts of evaluating GNNs, most benchmarks emphasize node classification accuracy, scalability, and computational efficiency without delving into the behavioral characteristics of GNNs in the spectral domain. This hinders a spectral understanding of the strengths and weaknesses of popular GNNs. Different from these benchmarks, we have proposed a comprehensive benchmark to evaluate both spatial and spectral GNNs from a consistent spectral view. To the best of our knowledge, it is a first-of-its-kind work that reveals insights challenging prevalent opinions from a spectral perspective.

## 7 CONCLUSION

This paper presents a comprehensive benchmark for evaluating GNNs from a spectral perspective, challenging common opinions about how GNNs process graph data. Through exploratory studies and rigorous experiments on real-world datasets, we demonstrate that GNNs can flexibly generate outputs with diverse frequency components, even when certain frequencies are absent in the input. This finding contradicts the prevailing opinion that neighborhood aggregation mechanisms dominate GNN behavior in the spectral domain. Our novel evaluation protocol, supported by theoretical analysis, provides a fundamental framework to measure the capability of GNNs to capture and leverage information across different frequency components. These findings open new avenues for understanding and improving GNN architectures, emphasizing the need to analyze GNNs in their entirety rather than focusing solely on their aggregation mechanisms. Our work lays the foundation for future research in spectral analyses of GNNs and provides valuable tools for practitioners to select and design more effective GNN models for various graph-based learning tasks.

## 8    ACKNOWLEDGMENTS

This work is supported in part by the National Science Foundation (NSF) under grants IIS-2006844, IIS-2144209, IIS-2223769, CNS-2154962, BCS-2228534, and CMMI-2411248; the Office of Naval Research (ONR) under grant N000142412636; the Commonwealth Cyber Initiative (CCI) under grant VV-1Q24-011; the UVA School of Engineering and Applied Science (SEAS) Research Innovation Award; the UVA Comprehensive Cancer Center (CCC) Computational Genomics and Data Science Pilot Award; and research gift funding from Netflix and Snap.

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

## A    REPRODUCIBILITY

All experiments were conducted using Python with the PyTorch (Paszke et al., 2017) and PyTorch Geometric (Fey & Lenssen, 2019) libraries. We detail the exact experimental settings for the preliminary study and main benchmark below. We also list all of our experimental hyperparameters in Table 2. Implementations are at: `https://github.com/yushundong/Spectral-benchmark`.

### A.1    PRELIMINARY STUDY

The datasets for the preliminary study are constructed by first taking each graph and computing its eigenvectors based on the normalized graph Laplacian. We then sort the eigenvectors by eigenvalue in ascending order and bin them into 20 uniform-width bins. Each bin contains the mean of the eigenvectors whose corresponding eigenvalues falling into that bin. We then split the bins into 3 uniform intervals corresponding to low, medium, and high frequencies.

We then task a randomly initialized GCN and FA with predicting the mean of these low, medium, and high frequency eigenvectors using the mean-squared error (MSE) loss using the eigenvectors of the complementary ranges as features. For each setting (e.g. using low frequencies as features and high frequencies as targets), we split our constructed dataset using an 80%/20%/20% train/validation/test split under a transductive node regression setting where we randomly generate the train/validation/test masks. During training, we standardize the eigenvector targets, subtracting their mean and dividing by their variance, which is a common practice in machine learning to facilitate learning during gradient descent (Hastie et al., 2009). During evaluation, we restore the outputs of the GNN back to the original scale.

### A.2    MAIN BENCHMARK

For the main benchmark, we construct each dataset similar to the preliminary study by collecting the normalized Laplacian eigenvectors of each graph followed by binning. Next, as this benchmark is a node classificaiton task, each entry of each binned eigenvector is assigned a class corresponding to that which bin it falls into between -1 and 1. For example, if the number of provided classes is 5, and a given binned eigenvector's entry is 0.8, then that entry will be assigned a class label of 4, the final class label.

Next, we conduct experiments where, for every bin $b$ of eigenvectors, we train each GNN to predict the classes of each entry of $b$ using the features from the original dataset, e.g. BOW text features for Cora-Full. We train our models using the cross-entropy loss and use a random 60%/20%/20% train/validation/test split under a transductive node classification setting. We detail our hyperparameter settings for our benchmark below in Table 2.

## B    PROOFS

In order to support the solidity of our design of the graph node class labels matrix (NCL matrix), we claim that the energy distribution of an eigenvector of the graph Laplacian matrix is close to that of its corresponding NCL matrix. To verify this claim, we first define the *energy distribution field* of a node class label distribution.

**Definition B.1.**  (Energy Distribution Field) The energy distribution field, denoted as $\mathcal{F}_M$, of an NCL matrix $M \in \mathbb{R}^{n \times k}$ is the set of energy distributions of the unit vectors whose corresponding NCL matrix is $M$. In other words, $\mathcal{F}_M := \{e(v) \in \mathbb{R}^{n \times 1} | ||v||_2 = 1, \tau(v) = M\}$, where $e(\cdot)$ is the

| Hyperparameter | Preliminary study value | Main benchmark value |
|---|---|---|
| # Layers | 3 | 2 |
| Hidden Dimension Size | 64 | 64 |
| Learning Rate | 0.0002 | 0.001 |
| Number of Epochs | 2000 | 500 |
| Optimizer | Adam | Adam |
| Dropout Rate | 0.0 | 0.0 |
| Learning Rate Scheduler | Cosine annealing, $T_0 = 10$ | None |
| Initialization | He (Kaiming) | Default PyTorch |
| Early stopping | None | None |

Table 2: Hyperparameters for preliminary study and main benchmark

energy distribution function mapping a vector to its energy distribution in the graph spectrum field, $\tau(\cdot)$ is the function mapping unit vectors to their NCL matrices.

With the concept of an energy distribution field (EDF), we formulate the "closeness" between the energy distribution between the eigenvector $v$ of graph Laplacian and its corresponding NCL matrix $\tau(v)$ as $\max_{e_u \in \mathcal{F}_{\tau(v)}} ||e_u - e(v)||_2$. We take two steps to validate that the optimal value of the maximization problem is "small enough": *(i)* We prove that the energy distribution function is Lipschitz, which indicates that it is a "smooth" function where quantifying its function value variations is equivalent to quantifying its variable variations; *(ii)* We prove that the inverse image of the energy distribution function $e(\cdot)$ on any energy distribution field $\mathcal{F}_M$ is "small enough".

For the first step, since the energy distribution function is essentially a linear function followed by a vector normalization procedure, we can write the energy distribution function as $e(v) = \frac{(\mathbf{A}v) \odot (\mathbf{A}v)}{||\mathbf{A}v||^2}$ where $\mathbf{A}$ is an orthonormal matrix. We can show that the energy distribution function is Lipschitz:

**Theorem B.2.** *The energy distribution function* $e(v) = \frac{(\mathbf{A}v) \odot (\mathbf{A}v)}{||\mathbf{A}v||^2}$*, with* $\mathbf{A}$ *being orthonormal, is Lipschitz on unit sphere.*

*Proof.* Assume there are two unit vectors $v_1$ and $v_2$, then

$$
\begin{aligned}
||e(v_1) - e(v_2)||_2 &= ||\frac{(\mathbf{A}v_1) \odot (\mathbf{A}v_1)}{||\mathbf{A}v_1||^2} - \frac{(\mathbf{A}v_2) \odot (\mathbf{A}v_2)}{||\mathbf{A}v_2||^2}||_2, \\
&= ||(\mathbf{A}v_1) \odot (\mathbf{A}v_1) - (\mathbf{A}v_2) \odot (\mathbf{A}v_2)||_2, \\
&= ||(\mathbf{A}v_1 - \mathbf{A}v_2) \odot (\mathbf{A}v_1 + \mathbf{A}v_2)||_2, \\
&\leq 2||\mathbf{A}(v_1 - v_2)||_2, \\
&\leq 2||\mathbf{A}||_2||(v_1 - v_2)||_2, \\
&= 2||v_1 - v_2||_2.
\end{aligned}
\tag{1}
$$

Here, the second equation follows because $\mathbf{A}$ is an orthogonal matrix, which implies that $||\mathbf{A}v_1||_2 = ||\mathbf{A}v_2||_2 = 1$. The third equation can be verified easily with the definition of Hadamard multiplication. The first "$\leq$" is derived from the observation that any entry of $\mathbf{A}v_1$ and $\mathbf{A}v_2$ has an absolute value less or equal to one since those two vectors are unit vectors. The second "$\leq$" follows by the Cauchy-Schwarz inequality. Finally, the last equality follows from the fact that $||\mathbf{A}||_2 = 1$ since $\mathbf{A}$ is an orthonormal matrix. $\square$

For the second step, we prove that the inverse image of the energy distribution function $e(\cdot)$ on any EDF $\mathcal{F}_M$ is "small enough". Intuitively, we can calculate the area of $e^{-1}(\mathcal{F}_M)$ on the unit sphere and prove that the area is "small enough". However, since the calculation of the area is rather complex for a high-dimensional sphere, we instead prove that the variation of center angle in $e^{-1}(\mathcal{F}_M)$ is small enough. Specifically, we have the following theorem:

**Theorem B.3.** *Assume that we segment* $[-1, 1]$ *(i.e., the value field of any entry of an eigenvector of a graph Laplacian matrix) into $k$ intervals with identical lengths, as illustrated in Figure 6(a). Consider the Euclidean space of* $[-1, 1]^n$ *separated into $k^n$ identical hypercubes. We claim that each section of the unit sphere and a hypercube corresponds to a* $e^{-1}(\mathcal{F}_M)$ *for some NCL matrix $M$. We denote the maximum center angular variation of* $e^{-1}(\mathcal{F}_M)$ *among all NCL matrices $M$s as* $\theta_{\max}$*. For* $\forall \epsilon \geq 0$*, there exists a* $k_0 = 2(\frac{2}{\pi e^{\frac{1}{e}}})^{\frac{n}{2}}$*, when $k \geq k_0$, then $\theta_{\max} \leq \epsilon$.*

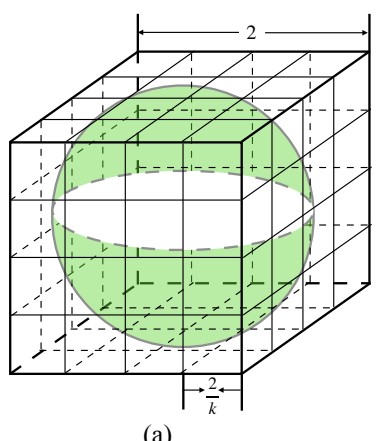 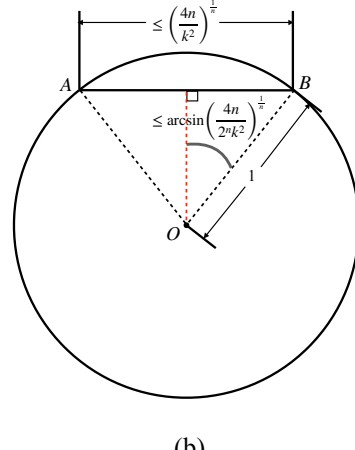

|         |         |
| :-----: | :-----: |
| (a)     | (b)     |

Figure 6: An illustration of key insights for the proof of Theorem B.4. (a) a unit sphere being sectioned by $k^n$ identical hypercubes, where each segment corresponds to a $e^{-1}(\mathcal{F}_M)$ for some NCL matrix $M$. (b) a 2-d tangent plane determined by point 0 and the chord with length smaller than or equal to $(n(\frac{2}{k})^2)^{1/n}$, where the angle of the chord forms is smaller than or equal to $2 \arcsin \frac{1}{2}(\frac{1}{n})^{\frac{1}{n}}(\frac{2}{k})^{\frac{2}{n}}$.

*Proof.* The largest chord length of a small hypercube sectioning the inscribed sphere is at most as long as the longest diagonal of the hypercube whose length is $(n(\frac{1}{k})^2)^{\frac{1}{n}}$. Therefore, we consider the 2D tangent plane determined by the point **0** and the chord with length smaller than or equal to $(n(\frac{2}{k})^2)^{\frac{1}{n}}$, as shown in Figure 6(b), the angle formed by **0** and the endpoints of the chord is smaller than or equal to $2 \arcsin \frac{1}{2}(\frac{1}{n})^{\frac{1}{n}}(\frac{2}{k})^{\frac{2}{n}}$. Hence, we have

$$\theta_{\max} \leq 2 \arcsin\left(\frac{1}{2}(\frac{1}{n})^{\frac{1}{n}}(\frac{2}{k})^{\frac{2}{n}}\right) \leq 2 \arcsin\left(\frac{1}{2}e^{\frac{1}{e}}(\frac{2}{k})^{\frac{2}{n}}\right) \leq \pi\left(\frac{1}{2}e^{\frac{1}{e}}(\frac{2}{k})^{\frac{2}{n}}\right). \tag{2}$$

Since we have $k \geq k_0 = 2(\frac{\pi e^{\frac{1}{e}}}{2})^{\frac{n}{2}}$, we conclude that $\theta_{\max} \leq \theta$. $\square$

Now, we are able to provide an upper bound for the energy distribution variance for EDF $\mathcal{F}_M$ of any NCL matrix $M$.

**Theorem B.4.** *Given the conditions of Theorem B.3, for any eigenvector $v$ of Laplacian amtrix of graph $\mathcal{G}$, we have $\max_{v \in \mathcal{S}^{n-1}, e_u \in \mathcal{F}_{\tau(v)}} ||e_u - e(v)||_2 \leq 2(\frac{4n}{k^2})^{\frac{1}{n}}$, where $\mathcal{S}^{n-1}$ is $n-1$ dimensional unit sphere embedded in $n$ dimensional Euclidean space.*

*Proof.* Since $e_u \in \mathcal{F}_{\tau(v)}$, then

$$\max_{v \in \mathcal{S}^{n-1}, e_u \in \mathcal{F}_{\tau(v)}} ||e_u - e(v)||_2 \leq \max_{v \in \mathcal{S}^{n-1}; e_u, e_l \in \mathcal{F}_{\tau(v)}} ||e_u - e_l||_2. \tag{3}$$

By Theorem B.3, we have

$$\max_{v \in \mathcal{S}^{n-1}; u, l \in e^{-1}(\mathcal{F}_{\tau(v)})} ||u - l||_2 \leq (\frac{4n}{k^2})^{\frac{1}{n}}. \tag{4}$$

With the Theorem B.2, we have

$$\max_{v \in \mathcal{S}^{n-1}; e_u, e_l \in \mathcal{F}_{\tau(v)}} ||e_u - e_l||_2 \leq 2 \max_{v \in \mathcal{S}^{n-1}; u, l \in e^{-1}(\mathcal{F}_{\tau(v)})} ||u - l||_2. \tag{5}$$

Therefore, we conclude that

$$\max_{v \in \mathcal{S}^{n-1}, e_u \in \mathcal{F}_{\tau(v)}} ||e_u - e(v)||_2 \leq 2(\frac{4n}{k^2})^{\frac{1}{n}}. \tag{6}$$

$\square$

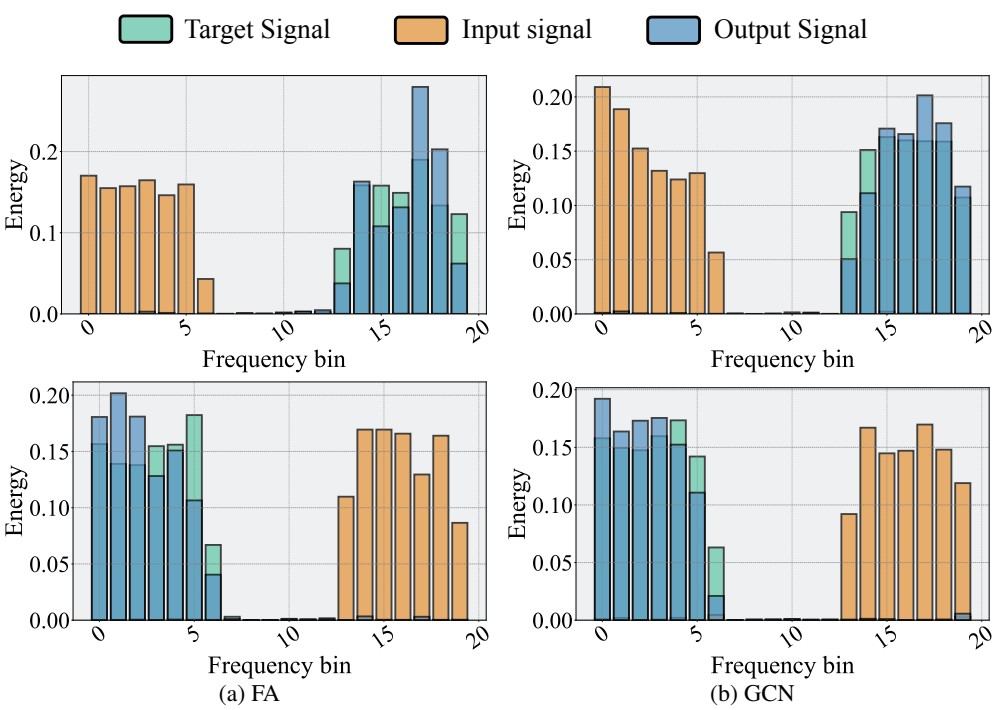

Figure 7: Energy distributions for CS.

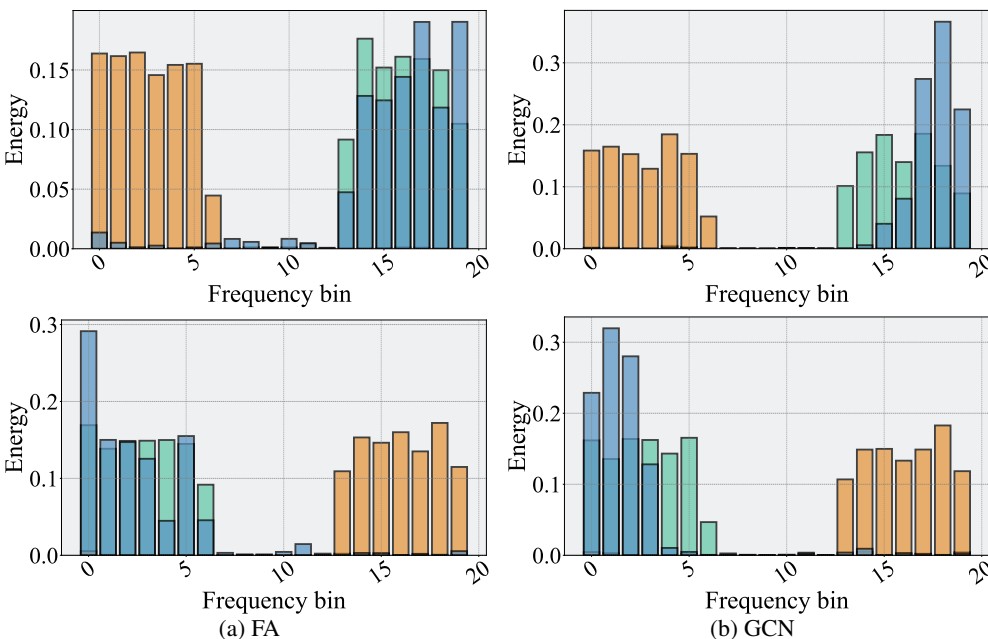

Figure 8: Energy distributions for Computers.

By Theorem B.4, given a tolerance of the change of energy distribution from an eigenvector and its corresponding NCL matrix, we can estimate the minimum number of identical interval segments $k$ in the design of NCL matrices to satisfy an arbitrary precision requirement.

## C    COMPLEMENTARY RESULTS OF EXPLORATORY STUDY

Figures 7, 8, 9, 10, 11 illustrates a comparison between the energy distributions of input and output frequency components of FA and GCN on the CS, Computers, Cora Full, Photo and Physics datasets.

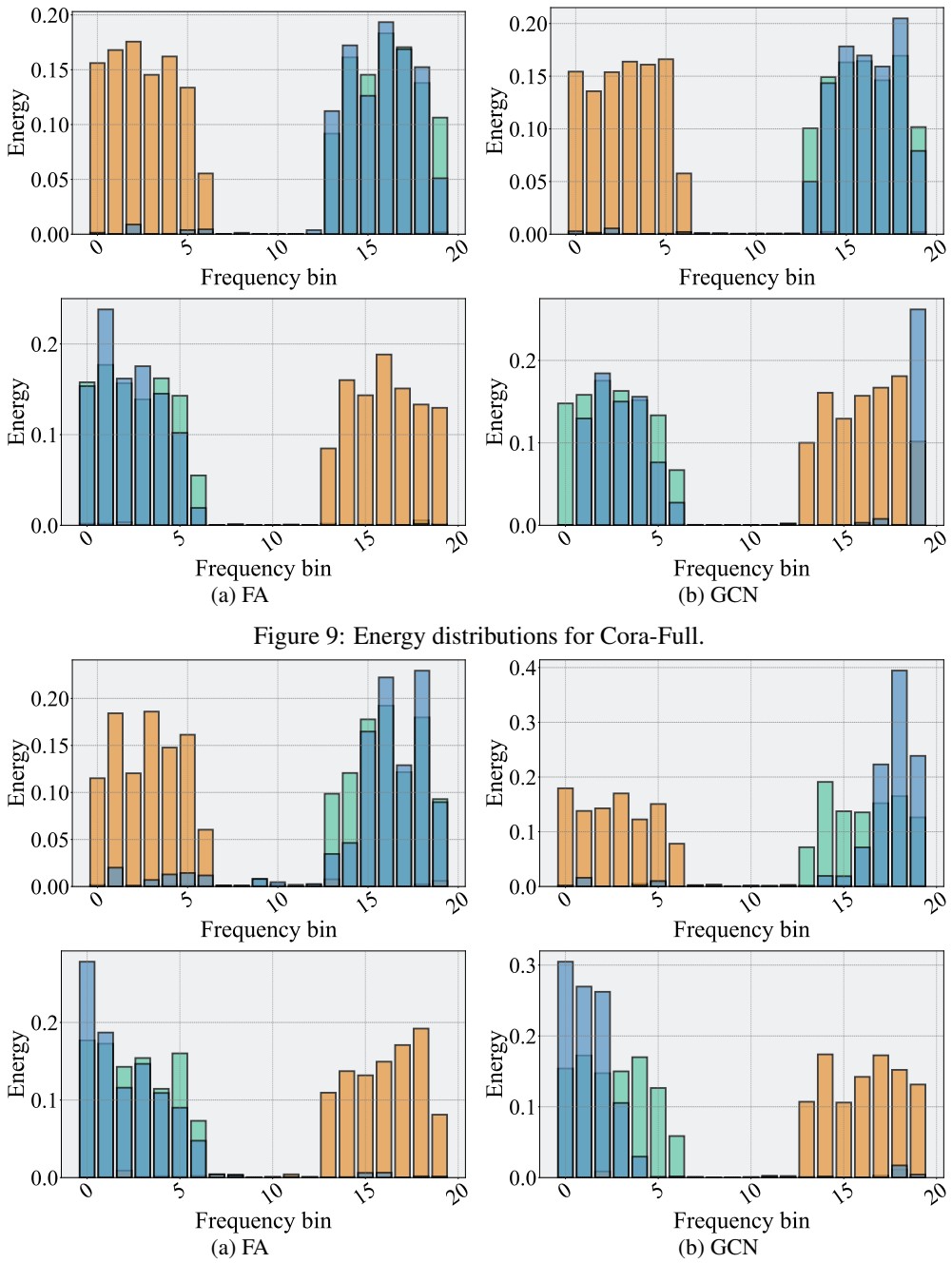

Figure 9: Energy distributions for Cora-Full.

Figure 10: Energy distributions for Amazon-Photo.

The plots depict the energy across 20 frequency bins, highlighting the distribution of input signals (orange), target signals (green), and output signals (blue).

We observe that FA and GCN performs similarly across all the examined datasets. Specifically, in each of those figures, the first subplot (top left) shows the FA model receiving an input concentrated in the lower frequency bins while the target signal is predominantly in the higher frequencies. Despite this discrepancy, the model's output aligns closely with the target frequency, successfully capturing the desired high-frequency components. Similarly, in the top right, the GCN model, which also starts with low-frequency input, produces an output that matches the high-frequency target distribution, demonstrating a similar flexibility. The lower two plots reverse the scenario: both models are now tasked with generating low-frequency outputs from high-frequency inputs. The FA model (bottom left) manages to shift the energy toward the low frequencies, although it retains some high-

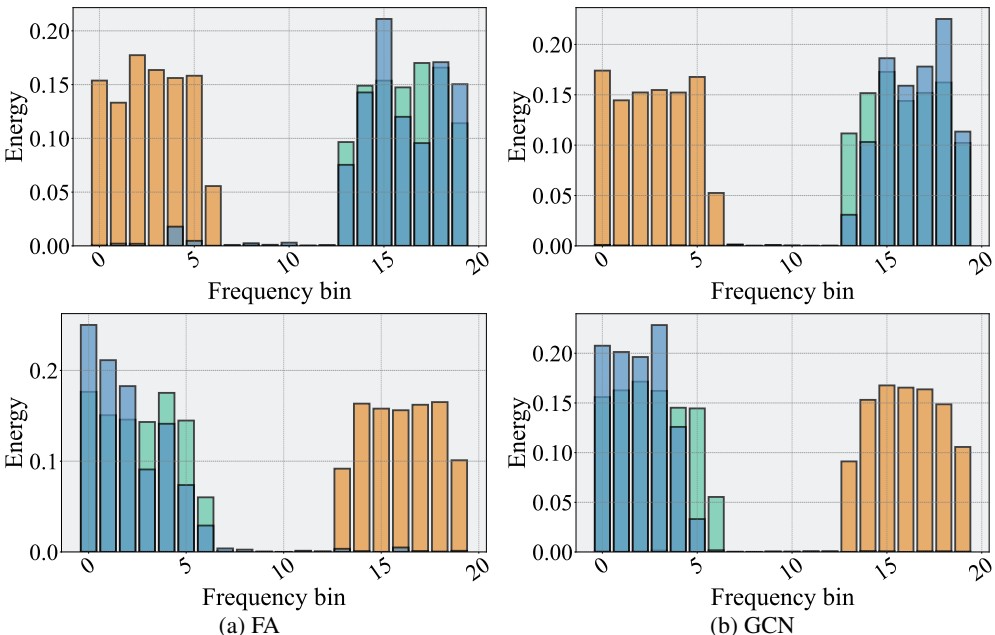

Figure 11: Energy distributions for Physics.

frequency characteristics from the input. Likewise, the GCN model (bottom right) shows a similar ability to recover low-frequency components, albeit with a smoother transition compared to FA.

These observations reveal that both models, FA and GCN, are capable of adjusting their output to align with the target distribution, even when the input and target signals differ significantly. This flexibility suggests that GNNs are not constrained solely by their neighborhood aggregation mechanisms in the spectral domain. Instead, other components, such as the non-linear layers in GNNs, play a crucial role in shaping the output, allowing the models to respond to frequency-specific incentives in the supervision signal.

## D COMPLEMENTARY RESULTS FOR QUALITATIVE PERFORMANCE

We show the complete version of Fig. 2 with more GNNs and datasets involved in Fig. 12 and Fig. 13. We have the observations as follows, which are consistent with the conclusion we draw from the Section 5.2. Generally, we observe that the V-shape curves are maintained across all GNNs in almost all datasets. The phenomenon indicates GNNs' strong ability in capturing the information encoded in the lowest and highest frequency components (e.g., those associate with the smallest and largest frequency values). We have provided a detailed rationale in Section 5.2. Note that we also detected a few cases such as 1-GNN on Photo and FA on DBLP which we consider to be outliers as their curves may be noisy in terms of accuracy variance, which hinders their frequency-capturing ability in the high frequency range. As expected, each GNN finds the most difficulty in recovering task-relevant information in the middle frequency components of each dataset.

## E COMPLIMENTARY RESULTS FOR QUANTITATIVE PERFORMANCE COMPARISON IN THE SPECTRAL DOMAIN

To more accurately study the performance of each GNN under our benchmark, we report the average GNN performance in capturing task-related information on different spectrum component areas. Specifically, Tables 3, 4, and 5 report average GNN performance in low, middle, high spectrum component areas. All metrics are multiplied by 100 for readability. We also **bolden** the highest metrics and underline the second-best metrics. Ties are broken by lower standard error.

In the *low-frequency range*, nearly all models perform well, but GatedGraph, GATv2, and GIN achieve particularly high accuracy, reflecting their strong low-pass filtering capabilities. For the *mid-frequency range*, ChebNet and GCNII demonstrate superior performance, while models like

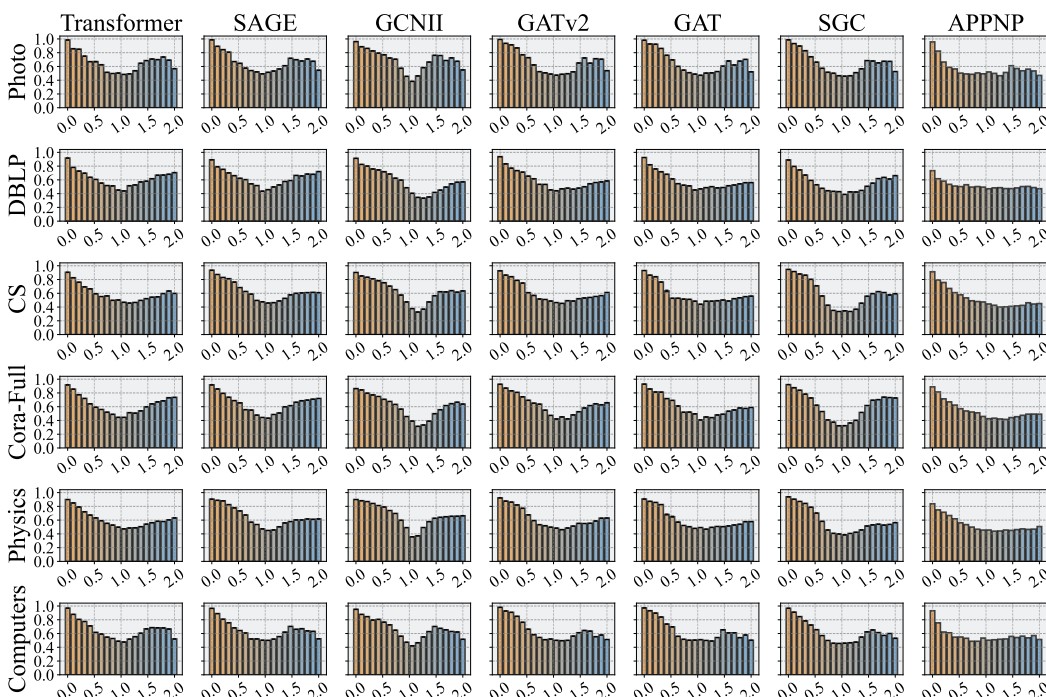

Figure 12: The accuracy curves of different GNNs (Transformer, SAGE, GNNII, GATv2, GAT, SGC, APPNP) in the whole spectral domain. In each subplot, the $x$-axis represents the frequency and the $y$-axis represents the accuracy of GNNs in the node classification task with the ground truth labels derived from the associated frequency bin.

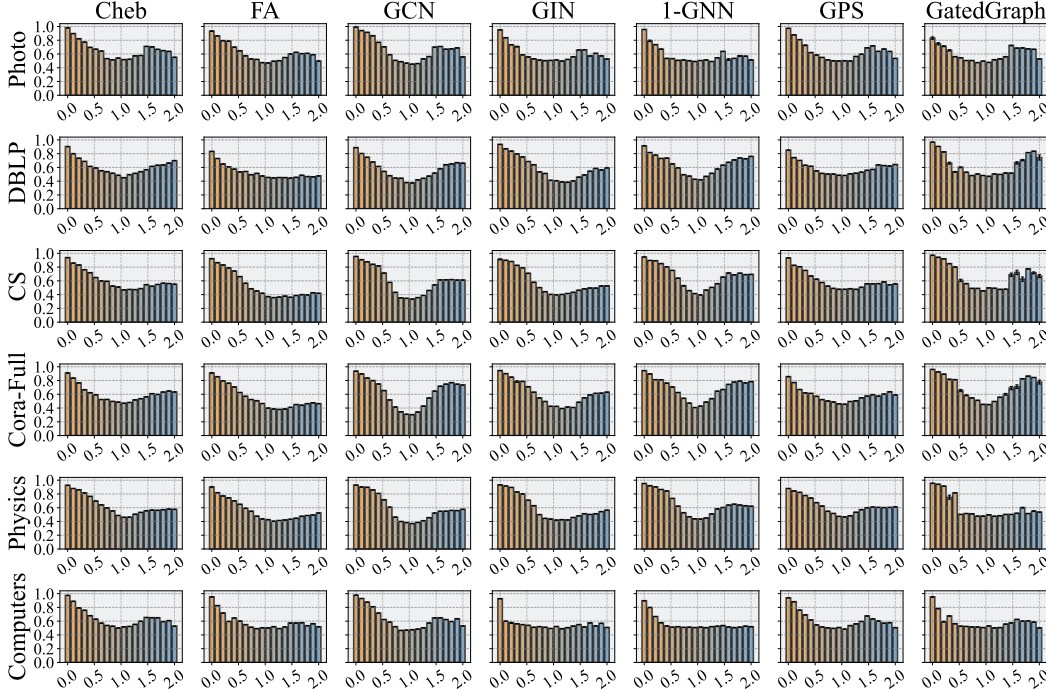

Figure 13: The accuracy curves of different GNNs (Cheb, FA, GCN, GIN, 1-GNN, GPS and GatedGraph) in the whole spectral domain. In each subplot, the $x$-axis represents the frequency and the $y$-axis represents the accuracy of GNNs in the node classification task with the ground truth labels derived from the associated frequency bin.

Table 3: AUC for the lower third of the spectrum (multiplied by 10)

| Model | Computers | Cora | CS | DBLP | Photo | Physics | Avg. Ranking |
|---|---|---|---|---|---|---|---|
| APPNP | 66.92 ± 2.22 | 71.35 ± 1.42 | 72.01 ± 1.58 | 57.99 ± 0.76 | 68.30 ± 3.02 | 69.10 ± 0.96 | 13.33 ± 0.75 |
| GPS | 73.89 ± 2.32 | 68.31 ± 1.19 | 76.81 ± 1.25 | 68.18 ± 1.14 | 76.52 ± 2.24 | 78.94 ± 0.56 | 11.17 ± 1.57 |
| FA | 72.27 ± 2.02 | 77.58 ± 1.04 | 80.16 ± 0.85 | 65.46 ± 1.21 | 78.70 ± 1.09 | 76.39 ± 0.79 | 10.17 ± 1.77 |
| Transformer | 79.24 ± 1.53 | 75.01 ± 1.54 | 74.07 ± 1.29 | 72.75 ± 1.25 | 79.70 ± 1.53 | 76.10 ± 1.08 | 9.83 ± 2.67 |
| Cheb | 78.66 ± 1.66 | 73.22 ± 1.58 | 79.22 ± 1.08 | 72.16 ± 1.37 | 80.59 ± 1.41 | 82.35 ± 0.69 | 8.83 ± 1.77 |
| GatedGraph | 68.20 ± 2.59 | **84.35 ± 1.21** | 84.86 ± 1.79 | 74.95 ± 3.05 | 67.55 ± 1.24 | 81.29 ± 2.92 | 7.50 ± 4.35 |
| SAGE | 79.10 ± 1.50 | 77.50 ± 1.01 | 81.61 ± 0.76 | 73.69 ± 0.95 | 80.81 ± 1.72 | 83.48 ± 0.46 | 7.17 ± 1.34 |
| GAT | 84.58 ± 1.20 | 80.34 ± 0.77 | 75.99 ± 2.33 | 75.26 ± 1.20 | 85.85 ± 1.19 | 79.89 ± 1.14 | 6.33 ± 3.59 |
| GIN | 62.51 ± 2.22 | 82.91 ± 0.74 | 83.61 ± 0.60 | **81.31 ± 0.77** | 72.92 ± 2.21 | 84.67 ± 0.71 | 6.17 ± 4.74 |
| SGC | 81.45 ± 1.36 | 79.45 ± 1.18 | 85.28 ± 0.74 | 70.34 ± 1.77 | 84.15 ± 1.50 | 84.09 ± 0.73 | 5.83 ± 2.73 |
| Graph | 66.35 ± 2.42 | 82.15 ± 0.75 | **85.67 ± 0.50** | 77.07 ± 0.79 | 70.36 ± 2.62 | **86.96 ± 0.58** | 5.67 ± 4.96 |
| GATv2 | **84.87 ± 1.38** | 81.19 ± 0.71 | 79.62 ± 1.22 | 77.35 ± 1.00 | **86.83 ± 1.03** | 82.15 ± 0.77 | 4.33 ± 3.14 |
| GCN | 82.20 ± 1.78 | 81.16 ± 1.04 | 85.14 ± 0.70 | 71.19 ± 1.60 | 86.40 ± 1.18 | 85.14 ± 0.62 | 4.33 ± 2.75 |
| GCNII | 84.03 ± 0.46 | 78.84 ± 0.35 | 82.22 ± 0.28 | 79.31 ± 0.49 | 85.00 ± 0.48 | 84.94 ± 0.18 | 4.33 ± 2.05 |

Table 4: AUC for the middle third of the spectrum (multiplied by 10)

| Model | Computers | Cora | CS | DBLP | Photo | Physics | Avg. Ranking |
|---|---|---|---|---|---|---|---|
| GCN | 50.01 ± 0.18 | 38.00 ± 0.60 | 39.81 ± 0.73 | 42.59 ± 0.14 | 49.91 ± 0.23 | 43.15 ± 0.69 | 13.33 ± 0.75 |
| SGC | 48.35 ± 0.17 | 38.90 ± 0.50 | 38.87 ± 0.67 | 43.13 ± 0.07 | 49.84 ± 0.16 | 43.69 ± 0.46 | 13.33 ± 0.47 |
| APPNP | 51.00 ± 0.04 | 47.29 ± 0.25 | 46.41 ± 0.20 | 49.46 ± 0.03 | 49.45 ± 0.03 | 47.04 ± 0.12 | 10.83 ± 1.57 |
| GIN | 50.75 ± 0.01 | 47.55 ± 0.75 | 44.88 ± 0.46 | 46.85 ± 0.93 | 51.16 ± 0.01 | 46.81 ± 0.58 | 10.83 ± 1.46 |
| FA | 50.96 ± 0.04 | 46.16 ± 0.57 | 43.05 ± 0.58 | 48.27 ± 0.12 | 50.89 ± 0.13 | 47.11 ± 0.53 | 10.83 ± 1.07 |
| Graph | 51.23 ± 0.00 | 49.92 ± 0.56 | 48.64 ± 0.66 | 48.48 ± 0.34 | 50.31 ± 0.01 | 49.30 ± 0.45 | 8.17 ± 2.03 |
| GatedGraph | 51.38 ± 0.01 | 51.41 ± 0.27 | 49.43 ± 0.10 | 49.57 ± 0.04 | 50.05 ± 0.04 | 49.09 ± 0.03 | 7.33 ± 2.75 |
| GAT | 51.19 ± 0.05 | 49.21 ± 0.45 | 49.28 ± 0.08 | 49.89 ± 0.10 | 51.81 ± 0.19 | 50.44 ± 0.11 | 7.17 ± 1.07 |
| GPS | 51.39 ± 0.07 | 48.97 ± 0.06 | 50.09 ± 0.14 | 50.15 ± 0.02 | 51.82 ± 0.07 | 52.05 ± 0.30 | 5.33 ± 1.37 |
| GATv2 | 52.20 ± 0.10 | **51.51 ± 0.96** | 49.90 ± 0.15 | 50.43 ± 0.37 | 51.39 ± 0.26 | 51.03 ± 0.19 | 4.50 ± 1.98 |
| GCNII | **54.93 ± 1.17** | 48.15 ± 2.10 | 49.69 ± 2.22 | 49.64 ± 1.99 | **55.50 ± 1.66** | **53.51 ± 2.23** | 4.17 ± 3.29 |
| Transformer | 52.92 ± 0.16 | 49.68 ± 0.17 | 50.10 ± 0.17 | 50.22 ± 0.17 | 52.21 ± 0.24 | 51.65 ± 0.19 | 4.17 ± 0.69 |
| SAGE | 53.24 ± 0.14 | 49.37 ± 0.23 | 51.40 ± 0.44 | **51.16 ± 0.31** | 53.45 ± 0.08 | 52.31 ± 0.63 | 3.00 ± 1.53 |
| Cheb | 53.25 ± 0.06 | 49.97 ± 0.05 | **52.06 ± 0.31** | 50.70 ± 0.11 | 54.92 ± 0.20 | 52.96 ± 0.49 | 2.00 ± 0.58 |

Table 5: AUC for the upper third of the spectrum (multiplied by 10)

| Model | Computers | Cora | CS | DBLP | Photo | Physics | Avg. Ranking |
|---|---|---|---|---|---|---|---|
| FA | 55.00 ± 0.07 | 44.14 ± 0.10 | 39.53 ± 0.05 | 46.36 ± 0.02 | 58.19 ± 0.20 | 46.85 ± 0.13 | 13.17 ± 1.21 |
| APPNP | 54.34 ± 0.05 | 46.59 ± 0.09 | 43.01 ± 0.05 | 48.53 ± 0.02 | 54.21 ± 0.20 | 46.87 ± 0.03 | 12.83 ± 0.69 |
| GIN | 53.92 ± 0.07 | 55.60 ± 0.68 | 48.87 ± 0.11 | 51.99 ± 0.50 | 59.39 ± 0.26 | 51.05 ± 0.13 | 11.67 ± 0.75 |
| GAT | 57.66 ± 0.27 | 54.30 ± 0.19 | 52.10 ± 0.08 | 52.50 ± 0.10 | 61.90 ± 0.54 | 53.60 ± 0.10 | 10.33 ± 0.94 |
| SGC | 59.17 ± 0.19 | 67.54 ± 0.66 | 57.33 ± 0.32 | 57.60 ± 0.61 | 63.67 ± 0.41 | 52.44 ± 0.11 | 7.50 ± 2.22 |
| GATv2 | 58.54 ± 0.22 | 59.04 ± 0.42 | 54.45 ± 0.15 | 52.92 ± 0.22 | 64.30 ± 0.68 | 57.34 ± 0.19 | 7.50 ± 1.50 |
| GPS | 59.38 ± 0.28 | 58.83 ± 0.08 | 55.20 ± 0.06 | 59.79 ± 0.20 | 64.09 ± 0.37 | 60.00 ± 0.02 | 6.83 ± 1.95 |
| Cheb | 61.12 ± 0.20 | 60.25 ± 0.16 | 53.97 ± 0.07 | 62.13 ± 0.30 | 64.24 ± 0.35 | 56.31 ± 0.03 | 6.67 ± 1.89 |
| Graph | 52.01 ± 0.01 | 73.94 ± 0.35 | **67.27 ± 0.29** | **68.78 ± 0.42** | 55.58 ± 0.18 | 62.32 ± 0.05 | 5.50 ± 5.68 |
| GCNII | 63.25 ± 0.35 | 57.26 ± 0.98 | 59.58 ± 0.35 | 48.46 ± 0.68 | **68.86 ± 0.53** | **63.84 ± 0.09** | 5.17 ± 4.63 |
| GCN | 60.72 ± 0.19 | 70.06 ± 0.62 | 58.13 ± 0.36 | 59.84 ± 0.60 | 65.17 ± 0.41 | 53.84 ± 0.16 | 5.17 ± 1.57 |
| GatedGraph | 57.80 ± 0.16 | **75.86 ± 0.92** | 66.80 ± 0.91 | 68.59 ± 1.66 | 64.67 ± 0.53 | 53.22 ± 0.12 | 4.83 ± 3.53 |
| Transformer | **64.40 ± 0.37** | 65.71 ± 0.50 | 56.23 ± 0.23 | 64.16 ± 0.28 | 67.64 ± 0.30 | 57.22 ± 0.17 | 4.33 ± 2.21 |
| SAGE | 63.54 ± 0.32 | 66.98 ± 0.22 | 59.13 ± 0.10 | 65.33 ± 0.27 | 66.22 ± 0.39 | 59.82 ± 0.05 | 3.50 ± 0.96 |

GATv2 and Transformer also perform consistently across multiple datasets. However, surprisingly, models such as FA and APPNP begin to show weaknesses in this region, with their performance dropping notably. In the *high-frequency domain*, GCNII once again leads the pack in Photo and Physics as well as the 1-WL graph operator while models such as GAT and FA struggle to maintain competitive results. Interestingly, SGC performs better than expected in the upper spectrum, despite being designed as a simplified GCN variant.

Overall, the trend shows that while most models are capable in the lower spectral ranges, only a few—like GCNII and Graph—exhibit balanced performance across the entire spectrum, particularly excelling in adapting to higher frequencies.

## F  COMPLEMENTARY RESULTS FOR PARAMETER STUDY

To demonstrate the robustness of our results, we run experiments under our benchmark settings using wider and deeper GNNs. Specifically, we re-run our benchmark using twice as many hidden dimensions (128 versus 64) and varying layer depths (up to 4) in order to test whether these hyperparameters can significantly affect each GNN's frequency adaptation abilities. In Figure 14, we present parameter studies across co-author datasets. In all of our ablations, we observe no significant shift in overall frequency adaptation behavior from the original plots in 2, suggesting that layer depth and width are not enough to improve or worsen GNN frequency adaptation capabilities.

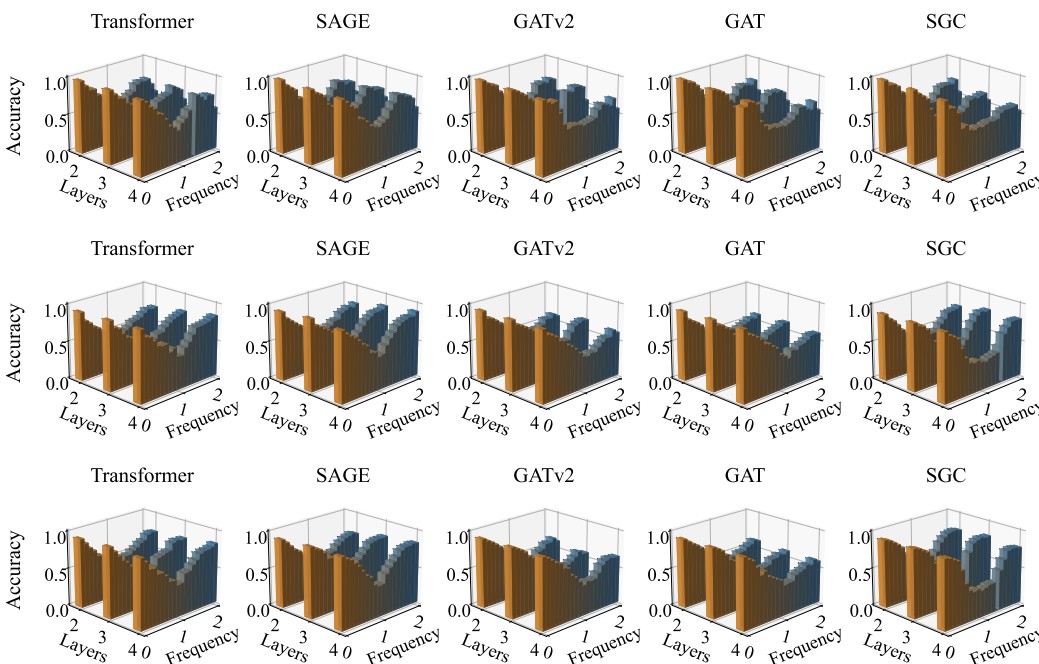

Figure 14: Parameter studies for Photo (top row), DBLP (moddle row) and Cora_full (bottom row), where the each subplot is a GNN's accuracy spectral tendency curve on a certain graph dataset with various (2 / 3 / 4) GNN layers .

