# OpenReview forum: "Graph Neural Networks Are More Than Filters: Revisiting and Benchmarking from A Spectral Perspective"
_ICLR.cc/2025/Conference — ICLR 2025 Poster_

### Official Review · Reviewer_oUdo · 2024-10-29

**Soundness:** 3
**Presentation:** 4
**Contribution:** 2
**Rating:** 6
**Confidence:** 5

**Summary:**

This work focusses on benchmarking current GNNs concerning their spectral perspective. It first conducts an exploratory study to show that GNNs can predict unseen frequency from the graph data and such ability is then evaluated in the protocol along with the node classification performance against 14 GNNs. The framework to measure the GNNs’ capability to capture and
leverage information across different frequency components is interesting.

**Strengths:**

1-	The evaluation from the spectral perspective is interesting and provides new insights.

2-	The presentation is good, making the paper easy to read.

**Weaknesses:**

1-	The EXPLORATORY STUDY needs more elaboration/evaluation. Current experiments show that GNNs can predict unseen frequency signals from graph data, which is not necessarily supports that “the filter resulted from the neighborhood aggregation does not dominate the behavioral characteristics of GNNs in the spectral domain”. Extra comparison with GNNs without graph signal input is needed to evaluate the extent of “A”.

2-	The scope of GNNs and datasets are restricted mostly to homophily ones. It would be helpful to incorporate heterophily GNNs considering their capability of high frequency domains.

3-	Lacks of discussion of existing spectral GNN benchmarks, for example [1].

[1] Benchmarking Spectral Graph Neural Networks: A Comprehensive Study on Effectiveness and Efficiency.

**Questions:**

In line 431, the author said “Instead, GNNs such as GATv2 and GCN achieve clear superiority over other GNNs in learning from middle frequency components.” Could you elaborate more why GCN is superior here? Upon middle frequency, the neighbors are often connected with mixed labels. GCN’s uniform aggregation can hardly distinguish such results.

I'm willing to raise the score if concerns can be well addressed.

---

> ### Author Response · Authors · 2024-11-18
> **Author Responses 1/2**
>
> We sincerely appreciate the time and effort you've dedicated to reviewing and providing invaluable feedback as well as recognizing the core intrigue of our work as providing new insights as a "framework to measure the GNNs' capability to capture and leverage information across different frequency components." **Such understanding aligns well with our core goal of benchmarking GNNs from a spectral perspective**. We provide a point-to-point reply below for the mentioned concerns and questions.
>
> ---
>
> > **Reviewer**: 1- The EXPLORATORY STUDY needs more elaboration/evaluation. Current experiments show that GNNs can predict unseen frequency signals from graph data, which is not necessarily supports that “the filter resulted from the neighborhood aggregation does not dominate the behavioral characteristics of GNNs in the spectral domain”. Extra comparison with GNNs without graph signal input is needed to evaluate the extent of “A”.
>
> **Authors**: We thank the reviewer for explaining this concern regarding the exploratory study. We agree with the reviewer that our expression here may lead to confusion. We would like to clarify our reasoning for our conclusion that "the filter resulted from the neighborhood aggregation does not dominate the behavioral characteristics of GNNs in the spectral domain" below.
>
> In graph signal processing (GSP), different graph filters correspond to different rewighting strategies on each of frequency component in the spectral domain. **Crucially, reweighting is an operation that is not able to recover missing frequency components**. Therefore, if a graph filter could have dominated GNN behavior in the spectral domain, we would expect GCN and FAGCN in our exploratory study to perform poorly when predicting unseen frequency signals since their aggregation schemes cannot recover missing frequency components. However, the results in Figure 1 demonstrates the opposite, indicating that the effect of each GNN's graph filter resulting from its neighobrhood aggregation scheme does not dominate its spectral capabilities, which supports our exploratory study's conclusion.
>
> We thank the reviewer for bringing up this concern, and we hope this clears any misunderstanding and more clearly states our argument.
>
> ---
>
> > **Reviewer**: 2- The scope of GNNs and datasets are restricted mostly to homophily ones. It would be helpful to incorporate heterophily GNNs considering their capability of high frequency domains.
>
> **Authors**: We thank the reviewer for pointing this out, and we would like to clarify the potential misunderstanding here regarding both GNNs and datasets below.
>
> Regarding GNNs, we agree that benchmarking with more GNNs built specifically for heterophilic graphs would improve the comprehensiveness of our study. We would like to note that our benchmark **already includes APPNP [1], GCNII [2], and FAGCN [3]**, which are all models with mechanisms built for highly heterophilic graph datasets with techniques such as averaging smoothed node embeddings with initial original node features (APPNP and GCNII) and using an adaptive filter for integrating diverse graph signals (FAGCN).
>
> Regarding heterophilic datasets, we agree with the reviewer that our benchmarking includes six homophilic graphs. However, we would like to highlight that **we also involved six heterophilic graph datasets in our case study**. With such a design, the satisfying estimation performance in determining which GNNs can perform better than others based on our benchmark validates the generalizability of our proposed evaluation strategy, indicating that the conclusions based on our benchmark do not rely on any homophily/heterophily assumption.
>
> We hope our discussion above helps address your concerns.
>
> [1] Predict then Propagate: Graph Neural Networks meet Personalized PageRank. Gasteiger et al. ICLR 2019.
> [2] Simple and Deep Graph Convolutional Networks. Chen et al. ICML 2020.
> [3] Beyond Low-frequency Information in Graph Convolutional Networks. Bo et al. AAAI 2021.

---

> ### Author Response · Authors · 2024-11-18
> **Author Responses 2/2**
>
> ---
>
> > **Reviewer**: 3- Lacks of discussion of existing spectral GNN benchmarks, for example [1].
> >
> > [1] Benchmarking Spectral Graph Neural Networks: A Comprehensive Study on Effectiveness and Efficiency.
>
> **Authors**: We thank the reviewer for referring us to this benchmark work, and we agree with the reviewer that discussing how our study differs from these existing ones will further improve the quality of our manuscript. We would like to note that the key difference between our work and [1] is that [1] mainly focuses on the **performance and efficiency** of spectral GNNs **rather than the flexibility of GNNs in producing different frequency components**, which is considered an evaluation with greater granularity. Specifically, our work distinguishes itself in two key ways:
>
> 1. Unlike [1], which emphasizes classification accuracy and efficiency, our focus lies in uncovering the **flexible behavior** of spectral GNNs in generating diverse frequency outputs. This includes the capability of spectral GNNs to adapt even when certain frequencies are absent or filtered from the input. Through this approach, we offer **exploratory studies and theoretical insights** that delve deeper into how GNNs process graph data and achieve satisfying performance from a spectral perspective, which [1] fails to present.
> 2. [1] primarily focuses on implementing and benchmarking existing spectral GNNs in a straightforward way by evaluating their performance on downstream tasks. However, we go beyond a straightforward evaluation by introducing a **novel evaluation strategy** grounded in theoretical analysis of energy distributions in the spectral domain. Such a strategy leads to comprehensive benchmarks that can scrutinize the performance of popular GNNs from a spectral perspective. Furthermore, the conclusion from such a benchmark can be directly used to **predict which GNNs can perform better in downstream tasks than other GNNs on new datasets**, which is also **first-of-its-kind** among existing benchmarks.
>
> These key points differentiate our work from others and also challenge the traditional perspectives of understanding the behavior of GNNs in the spectral domain. We believe that our work has the potential to open new avenues for research work investigating GNNs from a spectral perspective. We hope our supplementary discussion above helps address your concerns.
>
> ---
>
> > **Reviewer**: In line 431, the author said “Instead, GNNs such as GATv2 and GCN achieve clear superiority over other GNNs in learning from middle frequency components.” Could you elaborate more why GCN is superior here? Upon middle frequency, the neighbors are often connected with mixed labels. GCN’s uniform aggregation can hardly distinguish such results.
>
> **Authors**: We thank the reviewer for the insightful question regarding the performance of GCN, and we agree with the reviewer that this is an interesting phenomenon that challenges the traditional opinion of characterizing the superiority among popular GNNs. In middle-frequency scenarios, it is difficult for most GNNs to correctly distinguish nodes. However, we note that GCN does not really aggregate information uniformly -- the information is normalized symmetrically based on the degree of each node. As such, our preliminary hypothesis is that such a way of information aggregation implicitly improves learning the patterns encoded in the middle-frequency components, since degree information can be beneficial for predicting neighborhoods with mixed labels.
>
> However, we note that explaining why a particular GNN excels in certain frequency components is a difficult problem in general. The graph learning community has long been ignoring the comprehensive analysis of the performance of GNNs on different frequency components. As this work is an initial step towards a comprehensive analysis, **analyzing why this phenomenon occurs goes beyond the main focus of our paper**. Rigorously explaining such a phenomenon requires more in-depth explorations that we will leave to our future work. We hope our discussion above helps address your concerns.
>
> ---
>
> We thank you again for your valuable feedback on our work. With the further clarification, we believe that we have **responded to and addressed all your concerns with our point-to-point responses** — in light of this, **we hope you consider raising your score**. Please let us know in case there are any other concerns, and if so, we would be happy to respond.

---

> ### Author Response · Authors · 2024-11-25
> **A Kind Reminder**
>
> Dear Reviewer oUdo,
>
> Thank you for your valuable feedback on our work. We have prepared a thorough response to address your concerns. We believe that we have responded to and addressed all your concerns with our responses — in light of this, **we hope you consider raising your score as mentioned in the review**.
>
> Notably, given that we are approaching the deadline for the rebuttal phase, we hope we can receive your feedback soon. We thank you again for your efforts and suggestions!

---

### Official Review · Reviewer_EWWr · 2024-11-03

**Soundness:** 1
**Presentation:** 3
**Contribution:** 2
**Rating:** 3
**Confidence:** 4

**Summary:**

This paper introduces a benchmark designed to evaluate Graph Neural Networks (GNNs) from a spectral perspective. Through the benchmarking process, along with theoretical analysis and empirical evaluation, the authors aim to demonstrate that GNNs can flexibly produce outputs with a diverse range of frequency components, even when certain frequencies are missing in the input.

**Strengths:**

1.	The paper is well-structured.
2.	Many experiments are conducted.

**Weaknesses:**

1. Flawed assumptions in Experimental Design: The key assumption presented on line 183, which states, "We note that graph filters cannot generate frequency components that are not originally contained in the input graph data," appears flawed.

    * In a graph Laplacian space, high and low frequencies are interpreted relative to the eigenvalue spectrum rather than by fixed boundaries (which the authors also do not specify). For instance, even if all eigenvalues are small in magnitude, they can be ordered to produce relative high or low frequencies for graph filters. This explains why, in Figure 1, low-frequency inputs can recover high-frequency signals—because the model filters out relatively high-frequency portions within the low-frequency signal, which the neural network then amplifies to align with the high frequency target.

    * The performance using certain frequency as the training target cannot be used to explain the results produced by using label as the training target as they are not causally linked, and the theoretical analysis of the paper does not convincingly resolve this concern.

        * While models may learn high-frequency signals by training on low-frequency inputs, this does not guarantee that supervised learning with labels inherently captures the necessary high-frequency information. No evidence is provided to substantiate a causal relationship between certain frequency components and labels; either may be fitted independently. As noted on line 77, these ground-truth are usually a composition of different frequency components and does not show clear incentive patterns preferring certain frequency components.
        * While the theoretical analysis shows that discretizing a continuous energy distribution related to frequency has minimal deviation, it does not substantiate a fundamental connection between label distributions and discretized energy distribution. In other words, the discretization of energy distributions does not necessarily correlate with labels. The authors need to provide more detailed proofs to clarify a causal relationship between energy distribution and labels. However, given previous work [4][5], which shows that nodes of the same label on a heterophilic graph often present a mixed homophily of structures (demonstrating various frequencies), this causal link remains questionable.
2. Benchmark Design Limitations: On line 52, the authors mention that heterophilic graphs are more likely to benefit from high-frequency components in graph data. However, the benchmarking evaluation relies solely on homophilic graphs, which may inadequately capture GNNs' adaptive learning capabilities across different frequencies under label supervision. (Except for several heterophilic graphs only presented in case study).
3. Lack of Novel Insights in Experimental Results:
    * Qualitative Findings: Similar findings with explanations have been reported in existing works [2][3]. For example, traditional GCNs perform well in extreme low-frequency (high-homophily) and extreme high-frequency (low-homophily) conditions, struggling in mid-homophily settings. This is explained as low-homophily neighborhoods with sufficiently distinct averages can still benefit from low-frequency filtering.
    * Quantitative Performance Analysis Misrepresentation: The quantitative analysis lacks accuracy. In line 419, the authors claim that the spectral filtering ability of GCNII is “rarely discussed and evaluated,” yet the original study explicitly highlights its polynomial graph filtering capability. Additionally, since all datasets used are homophilic (dominated by low frequencies), this setup fails to provide sufficient evidence to support the claims made about GNN flexibility in frequency handling.
4.	Lack of Necessity in the Study's Focus:
    * Many GNNs are inherently capable of flexible filtering (e.g., GCNII and ChebNet used in benchmarking, both of which offer polynomial graph filtering), enabling them to encode a range of frequency components by design.
    * Theoretical results for GCNII also show that GNNs with initial residual connections can achieve polynomial graph filtering, implying that even standard GCNs can encode various frequencies with basic modifications such as residual connections. As shown in [1], classic GNNs can serve as strong baselines on heterophilic graphs requiring high-frequency information when paired with simple techniques like residual connections and dropout.

Reference
[1].	Luo, Yuankai, Lei Shi, and Xiao-Ming Wu. "Classic GNNs are Strong Baselines: Reassessing GNNs for Node Classification." arXiv preprint arXiv:2406.08993 (2024).

[2].	Ma, Yao, et al. "Is homophily a necessity for graph neural networks?." arXiv preprint arXiv:2106.06134 (2021).

[3].	Luan, Sitao, et al. "When do graph neural networks help with node classification? investigating the homophily principle on node distinguishability." Advances in Neural Information Processing Systems 36 (2024).

[4].	Yang, Liang, et al. "Diverse message passing for attribute with heterophily." Advances in Neural Information Processing Systems 34 (2021): 4751-4763.

[5].	Suresh, Susheel, et al. "Breaking the limit of graph neural networks by improving the assortativity of graphs with local mixing patterns." Proceedings of the 27th ACM SIGKDD conference on knowledge discovery & data mining. 2021.

**Questions:**

The meaning of "original task ranking" in Section 5.4 remains unclear. If node classification is the only task, does the benchmarking task ranking refer to an averaged ranking across multiple GNNs, with the original task ranking denoting the rank of a single GCN? A clearer explanation of this experimental setup would help clarify the significance of the conclusions presented in this section.

---

> ### Author Response · Authors · 2024-11-18
> **Author Responses 1/5**
>
> We sincerely appreciate your dedicated time and effort in reviewing and providing invaluable feedback. We provide a point-to-point reply below to clarify certain misunderstandings and provide responses to the mentioned concerns and questions.
>
> ---
>
> >  **Reviewer**: 1. Flawed assumptions in Experimental Design: The key assumption presented on line 183, which states, "We note that graph filters cannot generate frequency components that are not originally contained in the input graph data," appears flawed.
>
> **Authors**: We thank the reviewer for the feedback. We note that a widely acknowledged intuition of "filters" not only in the realm of graph signal processing (GSP) but also in the general field of signal processing is to re-weight the magnitude of different frequency components in the spectral domain. If the signal does not originally contain certain frequency components, only performing re-weighting cannot restore them.
>
> Below we present **a proof** to show why the claim **"graph filters cannot generate frequency components that are not originally contained in the input graph data"** holds in GSP.
>
> Traditionally, a filtering operation in GSP is defined as $\\mathbf{x} * g=\\boldsymbol{\\Phi} \\hat{g}(\\boldsymbol{\\Lambda}) \\boldsymbol{\\Phi}^T \\mathbf{x}=\\hat{g}(\\mathbf{L}) \\mathbf{x}$, where $\\mathbf{x} \\in \\mathbb{R}^{n \\times 1}$ is the node-level signal of the input graph (which can be considered as a column in the node feature matrix); $\\boldsymbol{\\Phi}$ and $\\boldsymbol{\\Phi}^T \\in \\mathbb{R}^{n \\times n}$ are the matrices formed by the eigenvectors of the graph Laplacian matrix $\\mathbf{L}$; $g$ denotes a graph filter. Here, $\\hat{g}(\\boldsymbol{\\Lambda})= diag\\left(\\hat{g}\\left(\\lambda_1\\right), \\hat{g}\\left(\\lambda_2\\right), \\ldots, \\hat{g}\\left(\\lambda_N\\right)\\right)$, and each $\\hat{g}\\left(\\lambda_i\\right)$ for $1 \\leq i \\leq N$ denotes the weight filter $g$ assigned to the frequency component at the frequency point $\\lambda_i$.
>
> When the input graph signal "does not contain certain frequency component" (e.g., the component at frequency point $\lambda_{k}$), then, by definition, $[\boldsymbol{\Phi}^T\mathbf{x}]_k=0$. After filtering, we calculate the magnitude of its frequency component at $\lambda_k$ as $[\boldsymbol{\Phi}^T\cdot(\mathbf{x}*g)]_k=[\hat{g}(\boldsymbol{\Lambda}) \boldsymbol{\Phi}^T \mathbf{x}]_k=\hat{g}\left(\lambda_k\right)[\boldsymbol{\Phi}^T\mathbf{x}]_k=0$. This means that no matter what graph filter is used in the filtering operation, if the graph signal does not originally contain certain frequency components (i.e., when certain elements in $\boldsymbol{\Phi}^T \mathbf{x}$ are zero), the graph filter cannot add them back. That is, $g$ cannot make the associated elements in the result $\hat{g}(\boldsymbol{\Lambda}) \boldsymbol{\Phi}^T \mathbf{x}$ become non-zero.
>
> We thank the reviewer for bringing this up, and we will also add the proof and the corresponding explanation to our paper to avoid misunderstanding. We hope this helps clarify your confusion.
>
> ---
>
> >  **Reviewer**: 1.1 In a graph Laplacian space, high and low frequencies are interpreted relative to the eigenvalue spectrum rather than by fixed boundaries (which the authors also do not specify). For instance, even if all eigenvalues are small in magnitude, they can be ordered to produce relative high or low frequencies for graph filters. This explains why, in Figure 1, low-frequency inputs can recover high-frequency signals—because the model filters out relatively high-frequency portions within the low-frequency signal, which the neural network then amplifies to align with the high frequency target.
>
> **Authors**: We thank the reviewer for the feedback, and we agree with the reviewer that all high and low frequencies are interpreted relative to the eigenvalue spectrum, which is defined by the Laplacian space. In our study, we have thoroughly tackled this issue: **we only compare the magnitude of high/low frequency in the same Laplacian space** by ensuring that the graph Laplacian being used is the same instead of using fixed boundaries. Therefore, we can ensure that for both input and output signals, small/large eigenvalues are always aligned to the same set of low/high-frequency points, respectively.
>
> We agree with the reviewer that Figure 1 may lead to the mentioned misunderstanding. We thank the reviewer for pointing this out, and we will elaborate on this discussion in our paper to avoid further misunderstanding.

---

> ### Author Response · Authors · 2024-11-18
> **Author Responses 2/5**
>
> ---
>
>
> >  **Reviewer**: 1.2 The performance using certain frequency as the training target cannot be used to explain the results produced by using label as the training target as they are not causally linked, and the theoretical analysis of the paper does not convincingly resolve this concern.
>
> **Authors**: We thank the reviewer for the feedback, and we would like to clarify a **misunderstanding** here: the conclusions presented in Section 5 **do not require the mentioned causal assumptions**. Instead, we aim to use our experiments to evaluate how well the output of GNNs can fit different types of signals across the spectral domain, which aligns with our research questions in Section 5.1.
>
> We further note that the theoretical analysis in Section 4 **does not aim to prove any causal relationship** between any involved variables. Instead, our goal is to derive Theorem 4.4 to show that node-level classification and regression tasks do not lead to significantly different output signals. Instead, the difference between their output signals is upper-bounded. We hope this clarifies the misunderstanding and addresses your concern.
>
> ---
>
>
> >  **Reviewer**: 1.2.1 While models may learn high-frequency signals by training on low-frequency inputs, this does not guarantee that supervised learning with labels inherently captures the necessary high-frequency information. No evidence is provided to substantiate a causal relationship between certain frequency components and labels; either may be fitted independently. As noted on line 77, these ground-truth are usually a composition of different frequency components and does not show clear incentive patterns preferring certain frequency components.
>
> **Authors**: We thank the reviewer for the feedback. We agree with the reviewer that "ground-truth are usually a composition of different frequency components and does not show clear incentive patterns preferring certain frequency components" as we mentioned in our paper. However, we note that the goal of our experiments is not to explore the causal relationship between certain frequency components and labels. Instead, we aim to explore the flexibility of different GNNs to produce different frequency components in their output under a pre-defined frequency incentive (when they are given the same input), which **does not require the mentioned causal assumption**. We hope this clarifies the misunderstanding and addresses your concern.
>
> ---
>
>
> >  **Reviewer**: 1.2.2 While the theoretical analysis shows that discretizing a continuous energy distribution related to frequency has minimal deviation, it does not substantiate a fundamental connection between label distributions and discretized energy distribution. In other words, the discretization of energy distributions does not necessarily correlate with labels. The authors need to provide more detailed proofs to clarify a causal relationship between energy distribution and labels. However, given previous work [4] [5], which shows that nodes of the same label on a heterophilic graph often present a mixed homophily of structures (demonstrating various frequencies), this causal link remains questionable.
>
> **Authors**: We thank the reviewer for the feedback, and we agree with the reviewer that our description at this step may be vague and can lead to misunderstanding. We note that our core goal is to evaluate the **capability of different GNNs to generate different frequency components flexibly**. To achieve this goal, the classifier at the end of the GNN model only serves as **a proxy of such performance evaluation by learning our binarization rule**. In this way, all frequency components encoded in the output of GNNs before input into the classifier remain continuous signals. Therefore, with the classifier serving as the performance proxy, we do not rely on any causal assumption between energy distribution and labels. We will revise the expression accordingly to ensure a rigorous description. We thank you again for bringing this up, and we hope the discussion above clarifies the misunderstanding.

---

> ### Author Response · Authors · 2024-11-18
> **Author Responses 3/5**
>
> ---
>
>
> >  **Reviewer**: 2. Benchmark Design Limitations: On line 52, the authors mention that heterophilic graphs are more likely to benefit from high-frequency components in graph data. However, the benchmarking evaluation relies solely on homophilic graphs, which may inadequately capture GNNs' adaptive learning capabilities across different frequencies under label supervision. (Except for several heterophilic graphs only presented in case study).
>
> **Authors**: We thank the reviewer for the feedback. We agree with the reviewer that six heterophilic graphs are used in benchmarking. However, we want to highlight that **we also adopt six highly heterophilic graphs in our case study**, where the ranking derived from our benchmark exhibits satisfying performance. Specifically, **such a design helps validate that our conclusion derived from our benchmarks does not rely on the homophily/heterophily assumption**. Rather, it reveals the robust generalization ability of our proposed evaluation strategy and further validates the practical significance of this study.
>
> Finally, we argue that performing empirical evaluation with **14 different GNNs** on **12 most representative datasets** (**six homophily** ones and **six heterophily** ones) can comprehensively validate the conclusions we presented in Section 5. We hope this helps address your concern.
>
> ---
>
>
> >  **Reviewer**: 3. Lack of Novel Insights in Experimental Results.
>
> **Authors**: We thank the reviewer for the feedback. We would like to summarize the novel insights from our study below:
>
> First, we **identify a significant issue of traditional GNN studies that has long been ignored**: graph filters cannot freely generate any frequency components needed as output, while GNNs can still achieve this. This means that such a capability comes from a factor other than graph filters, which is in conflict with the traditional understanding of GNNs. Evidently, this traditional understanding (i.e., analyzing only integrated filters) cannot completely account for GNN behavior in the spectral domain.
>
> Second, we propose **a novel evaluation strategy to understand the capability of GNNs as a whole to generate new frequency components**. On this basis, we obtain generalizable conclusions on the superiority of commonly used 14 popular GNNs even before any new experiments are performed on new datasets as verified by our case study.
>
> Third, our study **challenges the traditional way of investigating and understanding the performance and behaviors of GNNs in the spectral domain** and has the potential to recalibrate the research focus of the graph learning community onto this problem.

---

> ### Author Response · Authors · 2024-11-18
> **Author Responses 4/5**
>
> ---
>
>
> >  **Reviewer**: 3.1 Qualitative Findings: Similar findings with explanations have been reported in existing works [2] [3]. For example, traditional GCNs perform well in extreme low-frequency (high-homophily) and extreme high-frequency (low-homophily) conditions, struggling in mid-homophily settings. This is explained as low-homophily neighborhoods with sufficiently distinct averages can still benefit from low-frequency filtering.
>
> **Authors**: We thank the reviewer for the feedback. We would like to highlight that most existing works have discussed how low- and high-frequency components may influence the performance of GNNs. Although the GNNs' performance mid-frequency components have also been mentioned in several works, these existing works bear **two significant issues**.
>
> First, they **overwhelmingly analyze the frequency response function** but omit the neural networks in GNNs. This is understandable since obtaining the frequency response functions of neural networks is extremely difficult and may even be dataset-specific. However, this leaves the spectral behavior of GNNs as generally under-explored and poses a fascinating research question: do GNNs as a whole process different frequency components in exactly the same way as their integrated filters? To the best of our knowledge, no existing work has systematically explored this question.
>
> Second, what makes the situation worse is **the lack of strategy to explore how GNNs as a whole process different frequency components**. In fact, neither [2] nor [3] engaged in an in-depth discussion on understanding how mid-frequency components make a difference compared with low/high-frequency components. Specifically, [2] only performed empirical experiments on different datasets and did not explore the influence of different frequency components on each dataset. [3] merely mentioned that the mid-homophily (from the spatial domain) can be problematic, lacking an in-depth discussion from a spectral perspective. To the best of our knowledge, no existing work has proposed a comprehensive strategy investigating how each frequency component may be processed by different GNNs.
>
> In this work, we **surprisingly found that GNNs can process frequency components differently from their integrated filters**. Specifically, the traditional way of understanding GNNs, which focuses solely on analyzing their integrated filters, cannot account for GNN behavior in the spectral domain. This serves as a **novel insight** proposed for the first time, to the best of our knowledge. In addition, we proposed a **novel evaluation strategy** to gain a deeper understanding of such phenomena, and we derived generalizable conclusions that enable us to predict which GNN will perform better without the need to perform any specific downstream tasks. We hope this helps clarify the fundamental differences between our work and [2, 3] and addresses your concern.
>
> ---
>
>
> >  **Reviewer**: 3.2 Quantitative Performance Analysis Misrepresentation: The quantitative analysis lacks accuracy. In line 419, the authors claim that the spectral filtering ability of GCNII is “rarely discussed and evaluated,” yet the original study explicitly highlights its polynomial graph filtering capability. Additionally, since all datasets used are homophilic (dominated by low frequencies), this setup fails to provide sufficient evidence to support the claims made about GNN flexibility in frequency handling.
>
> **Authors**: We thank the reviewer for the feedback. We agree with the reviewer that our claim here is inaccurate. Our goal here is to highlight **the lack of in-depth discussion in the spectral domain for benchmarking these commonly used and representative GNNs**. We will revise the expression accordingly to ensure a rigorous description. We hope this helps address your concern.
>
> ---
>
> >  **Reviewer**: 4. Lack of Necessity in the Study's Focus.
>
> **Authors**: We thank the reviewer for the feedback. Our study is necessitated by the fact that most existing works overwhelmingly analyze the frequency response function but ignore analyzing GNNs as a whole. We **surprisingly found that GNNs can process frequency components differently from their integrated filters**. This reveals that the traditional understanding of GNNs which focuses on their integrated filters cannot faithfully reflect the behaviors of GNNs in the spectral domain. This reveals **a key issue that has long been ignored by the graph learning community**, and we hope this work can recalibrate the research focus on investigating GNNs as a whole in the spectral domain.

---

> ### Author Response · Authors · 2024-11-18
> **Author Responses 5/5**
>
> ---
>
> >  **Reviewer**: 4.1 Many GNNs are inherently capable of flexible filtering (e.g., GCNII and ChebNet used in benchmarking, both of which offer polynomial graph filtering), enabling them to encode a range of frequency components by design.
>
> **Authors**: We thank the reviewer for the feedback. We agree with the reviewer that many GNNs are inherently capable of flexible filtering. However, **this does not mean that they can freely generate any frequency components as their output**. As proved in our first response, graph filters cannot generate frequency components that are not originally contained in the input graph signal. Why these GNNs can still generate frequency components that are not originally contained in their input (proved in Section 3) is due to **a mechanism other than their integrated filters**. Such a mechanism **has been long ignored** by the graph learning community, and our work takes the first step to investigate this phenomenon with a carefully designed evaluation strategy and generalizable conclusions. We hope this helps address your concern.
>
> ---
>
>
> >  **Reviewer**: 4.2 Theoretical results for GCNII also show that GNNs with initial residual connections can achieve polynomial graph filtering, implying that even standard GCNs can encode various frequencies with basic modifications such as residual connections. As shown in [1], classic GNNs can serve as strong baselines on heterophilic graphs requiring high-frequency information when paired with simple techniques like residual connections and dropout.
>
> **Authors**: We thank the reviewer for the feedback. We agree with the reviewer that the performance of classic GNNs can be improved with simple techniques.
>
> We note that **this does not affect the contribution and novelty of our work**. The main goal of our work is to highlight the inherent flaw of analyzing GNNs by focusing exclusively on the frequency response functions of their integrated filters, and we take preliminary steps to handle this problem to gain a deeper understanding of GNNs' behaviors in the spectral domain. We hope this helps address your concern.
>
> ---
>
>
> >  **Reviewer**: Question: The meaning of "original task ranking" in Section 5.4 remains unclear. If node classification is the only task, does the benchmarking task ranking refer to an averaged ranking across multiple GNNs, with the original task ranking denoting the rank of a single GCN? A clearer explanation of this experimental setup would help clarify the significance of the conclusions presented in this section.
>
> **Authors**: We thank the reviewer for the feedback. We agree with the reviewer that the explanation here can be vague and may lead to misunderstanding. Here, we use the "original task ranking" to refer to the ranking of all GNNs under the node classification task on each dataset. Meanwhile, "Benchmark Ranking" denotes the ranking we collected from Figure 3. **The consistency between the two rankings implies that we can use the ranking in our benchmark to infer which GNNs can achieve better performance without needing to conduct empirical experiments on new datasets**.
>
> We will revise the expression accordingly to ensure rigorous description. We thank you again for bringing this up. We hope this helps address your concern.
>
> ---
>
> We thank you again for your valuable feedback on our work. With the further clarification, we believe that we have **responded to and addressed all your concerns with our point-to-point responses** — in light of this, **we hope you consider raising your score**. Please let us know in case there are any other concerns, and if so, we are eager to engage and make further clarifications.

---

> ### Author Response · Authors · 2024-11-25
> **A Kind Reminder**
>
> Dear Reviewer EWWr,
>
> Thank you for your valuable feedback on our work. We have prepared a thorough response to address your concerns. We believe that we have responded to and addressed all your concerns with our responses — in light of this, **we hope you consider raising your score**.
>
> Notably, given that we are approaching the deadline for the rebuttal phase, we hope we can receive your feedback soon. We thank you again for your efforts and suggestions!

---

> ### Author Response · Authors · 2024-11-27
>
> We are glad to share that we have achieved a consensus with all other three reviewers on their positive ratings, and we also appreciate your positive feedback on our experiments and presentation!
>
> However, considering that you have not replied to our carefully formulated point-to-point responses, we kindly reach out to request your feedback. We believe that we have responded to and addressed all your concerns, and hence we hope you consider raising your score.
>
> We thank you again for your efforts and suggestions, and we look forward to your feedback!

---

> ### Author Response · Authors · 2024-12-01
> **A Kind Reminder for the Last Two Days**
>
> We sincerely thank you for your efforts in helping us improve our paper! With only two days remaining and no response yet to our carefully formulated point-to-point replies, we are kindly reaching out to request your feedback. We believe we have thoroughly addressed all your concerns and respectfully hope you **consider raising your score**.
>
> Once again, we deeply appreciate your efforts and valuable suggestions, and we look forward to your feedback!

---

### Official Review · Reviewer_U2Hy · 2024-11-04

**Soundness:** 4
**Presentation:** 4
**Contribution:** 4
**Rating:** 6
**Confidence:** 2

**Summary:**

The paper shows that GNN is not only a neighbor's filter but can handle different types of graph information better than previously thought. This is interesting!

**Strengths:**

I really appreciate the thorough empirical work they did. Instead of just making claims, they:
did careful preliminary experiments to show this unexpected capability, tested on real-world datasets and made everything with theoretical analysis.

Another strength is how they formalized this into an evaluation framework. It's not just about showing this phenomenon - they developed systematic ways to measure and compare different GNNs' capabilities in handling various frequency components. This gives the field new tools to understand GNN behavior.

Their findings suggest we might need to rethink how we design and analyze GNNs. If GNNs aren't just filters but can actually generate new frequency patterns.

**Weaknesses:**

The benchmark may be overly simplified, as real-world tasks rarely exhibit such clear-cut frequency separations, limiting the applicability of the results to more complex, real-world scenarios.

Most experiments were done with 2-layer GNNs.

**Questions:**

Your findings suggest GNNs struggle with middle-frequency information. Could you elaborate on potential modifications or techniques that might specifically enhance GNN performance in this range? How can we evaluate it?


Have you validated the benchmark against any real-world applications (e.g., molecular biology, recommendation systems) to confirm its predictive capability in practical settings?

---

> ### Author Response · Authors · 2024-11-18
> **Author Responses 1/2**
>
> We sincerely appreciate your dedicated time and effort in reviewing and providing invaluable feedback. We also thank you for recognizing the novelty and the significance of our contributions. Plus, we believe that **your perspective** of "GNNs aren't just filters but can actually generate new frequency patterns" **aligns well with the core idea of this study**, **showing an in-depth understanding of our work**. We provide a point-to-point reply below for the mentioned concerns and questions.
>
> ---
>
> >  **Reviewer**: The benchmark may be overly simplified, as real-world tasks rarely exhibit such clear-cut frequency separations, limiting the applicability of the results to more complex, real-world scenarios.
>
> **Authors**: We thank the reviewer for the feedback. We agree with the reviewer that information may not exist following the clear-cut frequency separation patterns in real-world tasks. However, we note that for any dataset, **we can always manually obtain such frequency separations** and identify which frequencies bear the largest amount of energy — this is the only requirement to directly use our benchmark to estimate which GNN will be better for any new dataset.
>
> In fact, we follow such a strategy to conduct our case study (Section 5.4), which shows a satisfying estimation of the relative performance superiority between different GNNs **without the need to conduct any empirical experiments**. This reveals the practical significance of our novel work — determining which GNN will perform better on a new dataset and making appropriate choices prior to any empirical experiments.
>
> ---
>
> >  **Reviewer**: Most experiments were done with 2-layer GNNs.
>
> **Authors**: We thank the reviewer for the feedback. We would like to highlight that two-layer GNNs are **the most commonly used GNNs** in existing works due to their efficiency and competitive performance [1, 2, 3]. Therefore, adopting two-layer GNNs helps us to derive conclusions that can **directly benefit researchers and practitioners**. Meanwhile, our study also features a comprehensive selection of 14 popular GNN structures on 12 popular real-world datasets, which further helps ensure the wide applicability of our contributions.
>
> [1] Interaction-Aware Graph Neural Networks for Fault Diagnosis of Complex Industrial Processes. Chen et al. IEEE TNNLS, 2021.
>
> [2] Search to Capture Long-range Dependency with Stacking GNNs for Graph Classification. Wei et al. The Web Conference 2023.
>
> [3] Improving the Accuracy, Scalability, and Performance of Graph Neural Networks with ROC. Jia et al. MLSys 2023.
>
> ---
>
>
> >  **Reviewer**: Your findings suggest GNNs struggle with middle-frequency information. Could you elaborate on potential modifications or techniques that might specifically enhance GNN performance in this range? How can we evaluate it?
>
> **Authors**: We thank the reviewer for the feedback and for this question, which shows an in-depth understanding and a key insight into the intricacies of our paper. In fact, this brings up **a difficult problem that the graph learning community has been largely ignoring for a long time**: how does one make the best use of the information encoded in the middle-frequency components?
>
> Intuitively speaking, both the patterns of "a node tends to be similar to its neighbors" (low-frequency) and "a node tends to be dissimilar to its neighbors" (high-frequency) bring helpful information to the predictive power of GNNs, and thus are both beneficial for satisfying performance. However, when most energy falls in the middle-frequency components, the topology will exhibit a balance between the two cases above, which will confuse the GNN the most. To improve performance, we need to ensure that the GNNs can learn to exploit different patterns when the "usefulness" of these patterns varies with topology. **This is a difficult problem that goes beyond the main focus of our study**. We hope that our work can recalibrate the research focus of the graph learning community to investigate this difficult but critical problem, which also highlights the significance of our study.

---

> ### Author Response · Authors · 2024-11-18
> **Author Responses 2/2**
>
> ---
>
> >  **Reviewer**: Have you validated the benchmark against any real-world applications (e.g., molecular biology, recommendation systems) to confirm its predictive capability in practical settings?
>
> **Authors**: We sincerely thank the reviewer for the constructive feedback. We would like to highlight that the datasets and tasks utilized in our case study (Section 5.4) are derived from real-world applications across diverse domains, including public transportation and online web services. The strong performance of our proposed evaluation strategy demonstrates its capability to estimate which GNN can perform better for any new dataset we encounter. In accordance with your suggestions, we plan to further extend the application domains of our proposed evaluation strategy in our future work.
>
> ---
>
> We thank you again for your valuable feedback on our work. With the clarifications above, we believe that we have responded to and addressed all your concerns with our point-to-point responses — in light of this, **we hope you consider raising your score**. In the meantime, we believe that you have gained an in-depth understanding of our paper and our contributions. Hence we also hope you consider **raising your confidence**. Please let us know in case there are any other concerns, and if so, we would be happy to respond.

---

> ### Author Response · Authors · 2024-11-25
> **A Kind Reminder**
>
> Dear Reviewer U2Hy,
>
> Thank you for your valuable feedback on our work. We have prepared a thorough response to address your concerns. We believe that we have responded to and addressed all your concerns with our responses — in light of this, **we hope you consider raising your rating and confidence score**.
>
> Notably, given that we are approaching the deadline for the rebuttal phase, we hope we can receive your feedback soon. We thank you again for your efforts and suggestions!

---

> > ### Comment · Reviewer_U2Hy · 2024-11-26
> >
> > Thank you for your clarification and response. My concerns have been solved. Thanks!

---

> > > ### Author Response · Authors · 2024-12-01
> > >
> > > We are glad to hear that your concerns have been properly addressed, and we are grateful for your expertise and your dedicated efforts in helping improve our paper!

---

### Official Review · Reviewer_643X · 2024-11-04

**Soundness:** 3
**Presentation:** 3
**Contribution:** 3
**Rating:** 6
**Confidence:** 3

**Summary:**

This paper aims to study the effect of non-linear layers in graph neural networks (GNNs) on their spectral characteristics, in terms of their ability to represent information of different graph frequencies. Prior work focuses only on neighborhood aggregation and seeks to design filters that can represent graph frequencies more flexibly, but the current work considers GNNs holistically (including their nonlinearities) to empirically study their ability to represent different graph frequencies.

They first conduct a motivating experiment to show that GNNs can still recover frequency components which are filtered out by neighborhood aggregation, indicating the crucial role of nonlinear layers. Then, they construct a benchmark aiming to understand how well models can represent graph frequencies in different bucketized ranges. This is done by constructing classification tasks where the input is constructed using signals from one block of graph frequencies, and the output is a discretization of signals from a different block of graph frequencies, and then we measure the performance on this classification task. Theoretical results show that the discretization process does not give significant deviation in the spectral characteristics.

Then they empirically find using the benchmark that (1) GNNs show a U curve in their frequency response; (2) certain (mostly spatial) GNNs like SAGE perform better on this task; (3) the benchmark gives a useful ordering of methods that correlates with their performance on real-world (heterophilic?) node classification tasks; (4) the shape of the accuracy curves does not significantly change with the number of layers.

**Strengths:**

- Interesting and seemingly understudied problem aiming to consider GNNs holistically (including their nonlinearities) in the spectral domain.

- The accuracy curves seem like an interesting way of testing the ability of different GNNs to represent different frequencies, and provides an interesting "V curves" insight in RQ1.

- A real world case study is done, finding that the GNN rankings obtained from the benchmark are informative for deciding which GNNs are better than others on a series of real-world graph datasets.

**Weaknesses:**

- Clarity: some parts of the motivation are hard to understand, particularly due to some overly vague statements. For example, in the introduction, to make the motivation clear (particularly for readers who are less familiar with the cited works), I suggest precisely stating the common belief you are arguing against, and which papers exactly have demonstrated this common belief, and in what ways they use it. Similarly, Problem 2.1 is also too vague, e.g. it is unclear to the reader what "the key information" means. Similarly, in page 7, the paper also writes that RQ1 disagrees with "the prevalent opinion", without making it clear what this prevalent opinion is exactly, who expresses it, and why it is significant.

- The notion and computation of "accuracy curves" seems to be a main contribution of the paper, but I find it hard to appreciate what the reader should exactly take away from these accuracy curves, partly due to the complexity of their definition. From what I understand, they group the eigenvalues into (low, medium, high) frequencies, and then the mean of eigenvectors from 1 group to predict another group (as done in section 3), and then in section 4 it is the same except discretizing the outputs. So in figure 2, what are the input frequencies being used? Overall, it seems that the main goal of this section is to give the reader some useful insights, but it is hard to extract useful insights as the definitions are rather complex. In terms of insights, the V shape behavior in RQ1 is indeed somewhat interesting, but I still find that there is a lack of clearly practical insights.

- While the ranking of methods is also a practical insight in some sense, I am not quite convinced that this is really a practical way of ranking methods that one would actually want to use for real use cases. If I wanted to decide which GNN to use on a particular dataset, it seems more reasonable to rely on (1) the performance on the validation set or a held-out subset of the training set, or in the worst case (2) the performance on some other real-world datasets (which are ideally reasonably similar to mine). Does the current benchmark really perform better (or otherwise preferable in some way) at least compared to option (2) above (if so, it may be good to validate this assumption empirically)? Alternatively, if the authors could show that the benchmark provides some useful insights about methods (e.g. what causes SAGE to perform significantly better on this benchmark than other methods), that may also be useful.

- It is not clear how robust the rankings are to different choices of hyperparameters for the methods. Although the experiment in RQ4 does test the effect of different layer numbers, there is a lack of quantitative results (as the 3d plots are not clear enough to see to what extent the different hyperparameters actually affect the statistics such as normalized AUC or the ranking between methods).

Additional points (minor)

- Section 4.1: when describing the experimental protocol, please either describe all the necessary details, or cross-reference (e.g. to the appendix) to where they are given. For example, you should explain how the discretization is done (or cross-reference).

**Questions:**

- In the experiment in section 5.4, I could not understand what the authors mean by r4 being defined as "Original Task Ranking", or describing it as "the node classification rankings r4 directly derived from the node classification task on the chosen six datasets." In particular, how does this differ from r1 (which also seems to be the ranking of methods on the node classification task)?

- In the same experiment, please clarify if the datasets were specifically chosen as examples of heterophilic datasets (or if not, why these 6 datasets in particular were chosen).

- Figure 3 is somewhat confusing - the caption states that the 3 subfigures correspond to low, medium, and high frequency ranges, but then the x-axis of each subplot also corresponds to low, medium and high frequency ranges? Could you clarify what is the difference between these?

---

> ### Author Response · Authors · 2024-11-18
> **Author Responses 1/5**
>
> We sincerely appreciate the time and efforts you've dedicated to reviewing and providing invaluable feedback to enhance the quality of this paper. We provide a point-to-point reply below for the mentioned concerns and questions.
>
> ---
>
> >  **Reviewer**: Clarity: some parts of the motivation are hard to understand, particularly due to some overly vague statements. For example, in the introduction, to make the motivation clear (particularly for readers who are less familiar with the cited works), I suggest precisely stating the common belief you are arguing against, and which papers exactly have demonstrated this common belief, and in what ways they use it. Similarly, Problem 2.1 is also too vague, e.g. it is unclear to the reader what "the key information" means. Similarly, in page 7, the paper also writes that RQ1 disagrees with "the prevalent opinion", without making it clear what this prevalent opinion is exactly, who expresses it, and why it is significant.
>
> **Authors**: We thank you for pointing this out. We recognize the need to clarify the motivation of this work. To clarify, the common belief and "prevalent opinion" from page 7 we are arguing against is that the integrated graph filters induced by the aggregation schemes in GNNs are the dominating factor in determining the behavior of GNNs to process the input graph signal in the spectral domain. For example, [1] explains the performance of various GNNs by appealing to the strength of their graph filters. [2] argue that the polynomial filters of traditional GNNs are a major drawback, proposing a new GNN whose performance they attribute to its novel filter. The assumption behind this line of work is that each GNN's graph filter without its neural network is enough to dictate downstream performance. Our motivation in this work is to challenge this assumption by designing a strategy for characterizing **how GNNs can exhibit behavior that deviates from their integrated graph filters** in the spectral domain. As suggested, we will revise our statement here to clarify the discussion above.
>
> Regarding the phrase "key information", we agree that this phrase can be confusing and lead to misunderstandings. We want to clarify that "key information" in this context refers to the various low, mid, and high-frequency components of a given graph signal, respectively, in separate experiments. This is because they directly align with the ground truth information in our study by design.  We will revise Problem 2.1 to better clarify what we mean by "key information". We thank the reviewer again for their attentive reading and for pointing this out, and we hope our supplementary discussion helps address your concerns.
>
> [1] How Powerful are Spectral Graph Neural Networks. Wang and Zhang. ICML 2022.
> [2] Graph Neural Networks with Convolutional ARMA Filters. Bianchi et al. IEEE TPAMI, 2021.

---

> ### Author Response · Authors · 2024-11-18
> **Author Responses 2/5**
>
> ---
>
> >  **Reviewer**: The notion and computation of "accuracy curves" seems to be a main contribution of the paper, but I find it hard to appreciate what the reader should exactly take away from these accuracy curves, partly due to the complexity of their definition. From what I understand, they group the eigenvalues into (low, medium, high) frequencies, and then the mean of eigenvectors from 1 group to predict another group (as done in section 3), and then in section 4 it is the same except discretizing the outputs. So in figure 2, what are the input frequencies being used? Overall, it seems that the main goal of this section is to give the reader some useful insights, but it is hard to extract useful insights as the definitions are rather complex. In terms of insights, the V shape behavior in RQ1 is indeed somewhat interesting, but I still find that there is a lack of clearly practical insights.
>
> **Authors**: We thank the reviewer for the constructive feedback. We would like to first clarify the experimental setting of Figure 2 to explain its significance. Unlike the exploratory study in Section 3, the main benchmark in Section 4 does not take a certain group's binned eigenvectors as input. Rather, we directly **take the original graph dataset as the input**, and thus the GNNs will be fed with all frequency components contained in the original graph dataset (as explained in Appendix A.2). This is a more realistic scenario for real-world applications.
>
> With the discussion above, we would like to clarify further that each bar in Figure 2 generally characterizes **how capable a GNN is to output abundant frequency components within the associated bin of frequency**, since the frequency components from the associated bin are encoded as the supervision information for training by design (as explained in Appendix A.2). Therefore when a GNN can produce a bar close to 1 in all bins, this implies this GNN has strong capability to produce abundant frequency components within any bin of frequency. With such an understanding, there are two primary insights revealed by Figure 2.
>
> 1. While most works attribute poor performance on graphs with abundant high-frequency components to poor high-frequency filtering capabilities of the GNN's graph filter, we observe instead a V-shaped curve for many models, indicating that GNNs are actually able to go beyond their limited graph filter's capability to produce abundant high-frequency components.
> 2. Followingly, these accuracy curves demonstrate that contrary to popular belief, mid-frequency components are actually the hardest to leverage by GNNs. Their mix of lower and higher frequencies makes it difficult to learn a general criterion for node labels based on neighborhood information.
>
> These insights are the primary takeaways for Figure 2, and we will revise Section 5 to make these novel insights clearer. We thank the reviewer for the feedback, and we hope our discussion above helps address your concerns.

---

> ### Author Response · Authors · 2024-11-18
> **Author Responses 3/5**
>
> ---
>
> >  **Reviewer**: While the ranking of methods is also a practical insight in some sense, I am not quite convinced that this is really a practical way of ranking methods that one would actually want to use for real use cases. If I wanted to decide which GNN to use on a particular dataset, it seems more reasonable to rely on (1) the performance on the validation set or a held-out subset of the training set, or in the worst case (2) the performance on some other real-world datasets (which are ideally reasonably similar to mine). Does the current benchmark really perform better (or otherwise preferable in some way) at least compared to option (2) above (if so, it may be good to validate this assumption empirically)? Alternatively, if the authors could show that the benchmark provides some useful insights about methods (e.g. what causes SAGE to perform significantly better on this benchmark than other methods), that may also be useful.
>
> **Authors**: We thank the reviewer for explaining this concern. We agree that, in general, using a held-out validation set or comparison with similar real-world datasets is often helpful for ranking models and determining their relative effectiveness. However, we emphasize that ranking methods based on our benchmark has two key benefits: (1) the proposed approach is **better theoretically motivated** than most empirical methods for model selection; and (2) the proposed approach can be used to estimate the relative performance of a wide range of different GNN models **without the need of empirically trying each GNN** on a new dataset.
>
> For benefit 1, while using a held-out dataset or similar real-world dataset is simpler and straightforward, it **fails to leverage a fundamental understanding of the spectral behavior of GNNs** to make more generalizable decisions as to which models should perform better on certain datasets.
>
> For benefit 2, considering that there has been a wide range of popular GNN models to choose from, **empirically testing each GNN for every new dataset of interest can be computationally expensive**, especially in industrial scenarios where most datasets are large. Hence, an estimated approach prior to conducting any empirical experiments is valuable and aligns with the goal of our proposed evaluation approach (as demonstrated in our case study).
>
> We thank the reviewer for raising the question, and we hope our discussion above helps address your concerns.
>
> ---
>
> > **Reviewer**: It is not clear how robust the rankings are to different choices of hyperparameters for the methods. Although the experiment in RQ4 does test the effect of different layer numbers, there is a lack of quantitative results (as the 3d plots are not clear enough to see to what extent the different hyperparameters actually affect the statistics such as normalized AUC or the ranking between methods).
>
> **Authors**: We thank the reviewer for raising this concern, and we agree with the reviewer on the need to **provide quantitative results** in addition to the plots in Figure 5. As suggested, we put the quantitative results associated with Figure 5 measuring the normalized AUC (in percentage) of each model under each hyperparameter setting compared with the original benchmark setting below. Here $h$ refers to the hidden dimension size, and $L$ refers to the number of layers.
>
> | Model       | $h = 128, L = 2$ | $h = 128, L = 3$ | $h = 128, L = 4$ | $h = 64, L = 2$ (Table 1) |
> | ----------- | ---------------- | ---------------- | ---------------- | ------------------------- |
> | Transformer | 59.66 ± 1.56     | 61.28 ± 1.73     | 64.81 ± 1.73     | 59.43 ± 1.50              |
> | SAGE        | 62.66 ± 2.16     | 68.11 ± 2.36     | 70.36 ± 2.36     | 63.17 ± 2.01              |
> | GATv2       | 60.97 ± 2.07     | 59.60 ± 2.19     | 59.09 ± 2.41     | 60.41 ± 2.12              |
> | GAT         | 58.86 ± 2.11     | 57.07 ± 2.23     | 58.00 ± 2.03     | 58.28 ± 2.09              |
> | SGC         | 59.92 ± 4.41     | 60.47 ± 4.59     | 56.72 ± 4.07     | 59.25 ± 4.19              |
>
> From this table, we see that GNNs tend to retain similar relative rankings with each other even after increasing $h$ and $L$, with SAGE consistently outperforming competing methods and GAT consistently underperforming relative to the other GNNs. With this, the relative ranking of each GNN is not vastly affected by our choice of hyperparameters, validating the robustness of our benchmark. We also have **similar observations on other GNNs and hyperparameters**, and we will also **add comprehensive quantitative results to our Appendix** to further clarify the robustness of the ranking derived from our proposed evaluation approach.
>
> We thank the reviewer for pointing this out, and we will add the quantitative results together with the detailed discussion above to our paper. We hope this addresses your concern.

---

> ### Author Response · Authors · 2024-11-18
> **Author Responses 4/5**
>
> ---
>
> > **Reviewer**: Section 4.1: when describing the experimental protocol, please either describe all the necessary details, or cross-reference (e.g. to the appendix) to where they are given. For example, you should explain how the discretization is done (or cross-reference).
>
> **Authors**: We thank the reviewer for making this suggestion, and we agree that our introduction of the experimental protocol can lead to confusion. For clarity purposes, **we introduce the full experimental protocol in Appendices A.1 and A.2**. Meanwhile, the discretization process is also carefully described in Appendix A.2. We will revise this section accordingly to include more cross-references properly to avoid confusion. We thank you again for bringing this up, and we hope this helps address your concerns.
>
> ---
>
> > **Reviewer**: In the experiment in section 5.4, I could not understand what the authors mean by r4 being defined as "Original Task Ranking", or describing it as "the node classification rankings r4 directly derived from the node classification task on the chosen six datasets." In particular, how does this differ from r1 (which also seems to be the ranking of methods on the node classification task)?
>
> **Authors**: We thank the reviewer for pointing this out, and we understand that our explanation of each ranking method may lead to confusion. We would like to clarify the meaning of each ranking below.
>
> * $r_1$: This is the node classification performance ranking on the new datasets (AB, Wisc., etc.) on which we wish to perform node classification in the case study.
> * $r_2$: This is the ranking induced by node classification performance under our benchmark settings. This is a node classification task based on our evaluation strategy outlined in Section 4.1. Note that $r_2$ can be calculated based on the height of the bars in the lowest 1/3, middle 1/3, or highest 1/3 determined by which frequency range bears the most energy of each new dataset in Section 5.4.
> * $r_3$: This is a random ranking across the 14 GNNs.
> * $r_4$: This is the ranking induced by node classification performance on the **benchmark datasets under a traditional node classification task setting**. For example, for CS, this is 15-way node classification based on labels from the original Coauthor dataset from [1].
>
> In summary, $r_1$ is the node classification performance ranking on the new datasets (AB, Wisc., etc.), while $r_4$ is the node classification performance ranking on the benchmark datasets under their original task settings. **The main takeaway of our case study is that $r_2$ serves as a satisfying performance superiority estimation of $r_1$**. In other words, for any new dataset we are interested in, by checking $r_2$ presented in our benchmark, we can infer which GNNs enjoy better performance than others **without the need to conduct empirical experiments on such a new dataset**.
>
> We will revise our wording and explanation to make the definition of each ranking clear. We thank you again for pointing this out, and we hope this clears any confusion on the meaning of each ranking.
>
> [1] Pitfalls of Graph Neural Network Evaluation. Shchur et al. NeurIPS 2018.
>
> ---
>
> > **Reviewer**: In the same experiment, please clarify if the datasets were specifically chosen as examples of heterophilic datasets (or if not, why these 6 datasets in particular were chosen).
>
> **Authors**: We thank the reviewer for pointing this out. We clarify that these datasets were chosen as examples of heterophilic graph datasets which differ from those used in the benchmark by our design. Notably, we found that despite the significant differences of these new datasets (compared with the datasets in our benchmark), the performance ranking given by our benchmark still enjoys satisfying capability in estimating which GNNs perform better than others on these new datasets (i.e., the ranking $r_2$ serves as a satisfying performance superiority estimation of $r_1$). Therefore, our case study highlights the **generalizability and practical significance** of our conclusion in predicting downstream GNN performance based on their spectral behavior under our evaluation strategy.

---

> ### Author Response · Authors · 2024-11-18
> **Author Responses 5/5**
>
> ---
>
> > **Reviewer**: Figure 3 is somewhat confusing - the caption states that the 3 subfigures correspond to low, medium, and high frequency ranges, but then the x-axis of each subplot also corresponds to low, medium and high frequency ranges? Could you clarify what is the difference between these?
>
> **Authors**: We thank the reviewer for pointing this out, and we agree that this can lead to confusion. To clarify, Figure 3 **separates the 14 GNNs** that perform the best in low, medium, and high frequency ranges into three different subfigures **for better visualization purposes**. For example, SAGE only appears in the first subfigure. This means that SAGE is a GNN with the best performance in the low-frequency range, while the first subfigure also shows its performance in all three ranges to show how the performance changes across the three ranges. We will add a more detailed clarification about Figure 3, and we hope this helps clarify your confusion.
>
> ---
>
> We thank you again for your valuable feedback on our work. With our further clarification, we believe that we have **responded to and addressed all your concerns with our point-to-point responses** — in light of this, **we hope you consider raising your score**. Please let us know in case there are any other concerns, and if so, we would be happy to respond.

---

> > ### Comment · Reviewer_643X · 2024-11-22
> > **Reviewer's Response**
> >
> > Thanks for the replies; most of my issues are addressed. As a result, I am mostly willing to increase the score, but just have a few remaining points or questions.
> >
> > **Response to 2/5**: That makes sense to me now (and the overall intuition as well). The confusion came from Line 226 which says "Specifically, we propose to adopt the same approach to generate continuous values..." which might be interpreted as saying you adopt the same procedure as the previous section for generating both features and labels. Meanwhile, it is also better for the text to explicitly say that in this section, you use the features from the actual real graph.
> >
> > Just to clear a remaining doubt, the V-curve describes the pattern of accuracies as we vary the frequency; but I wonder if we can know whether that could be biased by differing label frequencies (for example, could it be that for low / high frequencies, the classification task is made easier due to relatively skewed label distributions)?
> >
> > I also wonder about the extent to which the benchmark results could be influenced by overfitting (which could vary based on the amount of training data) - for example, in table 1, GAT and GPS perform overall worse than GCN despite being generally considered more powerful models; do you think this is due to overfitting / lack of training data, or due to representation power reasons (i.e., differences in their ability to reconstruct different graph frequencies)?
> >
> > **Response to 5/5**: That makes sense. It would probably be clearer to say that you divide the GNNs into 3 groups; (1) those that perform better on "low" than "mid" and "high", (2) etc; then precisely write down which GNN is in each group; then label each subfigure with the group shown in it.

---

> > > ### Author Response · Authors · 2024-11-22
> > > **Authors' Response**
> > >
> > > We are glad to hear that most of the concerns have been properly addressed, and we are grateful for your dedicated efforts and expertise. Below we would like to clarify your remaining questions.
> > >
> > > ---
> > >
> > > **Regarding the response to 2/5:** We acknowledge that your understanding is correct. We appreciate your suggestions for revision, and we will revise our expression at line 226 for a clearer description.
> > >
> > > For the remaining doubt, it is an insightful perspective, and we acknowledge that differing label frequencies **may lead to different levels of difficulty** in achieving high-quality predictions. We have considered this effect, and we believe **such an effect should not be excluded** from generating the V-shaped curves. This is because such an effect is naturally coupled with varying the abundance of different frequency components in practice, and having this effect involved will ensure the practical significance of our benchmark. Therefore, your understanding is correct, and this will **not jeopardize the quality of the evaluation**.
> > >
> > > For the concern of overfitting, we acknowledge that we have utilized **a validation set** occupying 20% of the nodes for each experiment, and **only the optimized GNNs with the lowest validation loss** were selected to be evaluated on the test set. Meanwhile, the scales of the adopted graph datasets are also widely acknowledged to be able to provide satisfying evaluation performance. Therefore, the effect of overfitting has been **considered and carefully controlled** in our experiments.
> > >
> > > The last question also brings up **another exciting perspective**: GNNs with stronger expressiveness (e.g., GAT compared with GCN and SGC) do not necessarily enjoy a stronger capability to produce different frequency components flexibly. This may be attributed to their different ability to reconstruct different graph frequency components, but it still requires much further exploration. Note that this problem has long been ignored by most existing works, and our work has the potential to serve as an initial inspiration for this line of research.
> > >
> > > ---
> > >
> > > **Regarding the response to 5/5:** We thank the reviewer for the suggestion about revising the expression here. We will revise our manuscript as suggested to make the discussion clearer to further improve the quality of our work.
> > >
> > > ---
> > >
> > > With further clarification above, we believe your concerns can be fully addressed. In light of this, **we hope you consider raising your score**. Again, we appreciate your dedication to the review process and your efforts to help the quality of our work.

---

> > > > ### Comment · Reviewer_643X · 2024-11-22
> > > >
> > > > Thanks for the replies; my issues are satisfactorily addressed, so I have increased the score.

---

> > > > > ### Author Response · Authors · 2024-11-22
> > > > >
> > > > > We are glad to hear that all the concerns have been properly addressed, and we are grateful for your expertise and your dedicated efforts in helping improve our paper!

---

### Comment · Area_Chair_jSs1 · 2024-11-28

I would like to encourage the reviewers to engage with the author's replies if they have not already done so. At the very least, please
acknowledge that you have read the rebuttal.

---

### Meta-Review · Area_Chair_jSs1 · 2024-12-19

**Metareview:**

This paper introduces a benchmark to evaluate GNNs from a spectral perspective, aimed at their ability to represent diverse frequency components. It shows that GNNs, unlike standard graph filters, can recover filtered frequency components, highlighting the importance of the nonlinearity. The majority of the reviewers were in agreement that the results are insightful and interesting, and appreciated the thorough empirical work. The accuracy curves are well motivated, and the graph community will likley benefit from the insights in the paper, even though the real-world applicability of the rankings from the benchmark is not entirely clear.

Yes, it is interesting that "filters cannot generate frequency components that are not originally in the input, but GNNs can", but still it is fair to ask *why* or when is this ability important (for node classification on real-world graphs). Relatedly, I think some of the criticism from Reviewer EWWr is valid, even though I do not agree that it warrants a rejection. Investigation in the "causal" question that was brought up can be an intersting avenue for future work.

**Additional Comments On Reviewer Discussion:**

The authors were able to address the majority of the concerns raised by the reviewers, which also led some reviewers to raise the score. Two reviewers brought up a concern regarding homophilic / heterophilic datasets and the authors clarified that their evaluation is on 12 graphs (6 representatives from each). I encourage the authors to incorporate the constructive comments on presentation and structure.

---

### Decision · Program_Chairs · 2025-01-22

Accept (Poster)